# The Inductive Bias of Quantum Kernels

**Jonas M. Kübler**[*]    **Simon Buchholz**[*]    **Bernhard Schölkopf**
Max Planck Institute for Intelligent Systems
Tübingen, Germany
{jmkuebler, sbuchholz, bs}@tue.mpg.de

## Abstract

It has been hypothesized that quantum computers may lend themselves well to applications in machine learning. In the present work, we analyze function classes defined via *quantum kernels*. Quantum computers offer the possibility to efficiently compute inner products of exponentially large density operators that are classically hard to compute. However, having an exponentially large feature space renders the problem of generalization hard. Furthermore, being able to evaluate inner products in high dimensional spaces efficiently by itself does not guarantee a quantum advantage, as already classically tractable kernels can correspond to high- or infinite-dimensional reproducing kernel Hilbert spaces (RKHS).

We analyze the spectral properties of quantum kernels and find that we can expect an advantage if their RKHS is low dimensional and contains functions that are hard to compute classically. If the target function is known to lie in this class, this implies a quantum advantage, as the quantum computer can encode this *inductive bias*, whereas there is no classically efficient way to constrain the function class in the same way. However, we show that finding suitable quantum kernels is not easy because the kernel evaluation might require exponentially many measurements.

In conclusion, our message is a somewhat sobering one: we conjecture that quantum machine learning models can offer speed-ups only if we manage to encode knowledge about the problem at hand into quantum circuits, while encoding the same bias into a classical model would be hard. These situations may plausibly occur when learning on data generated by a quantum process, however, they appear to be harder to come by for classical datasets.

## 1    Introduction

In recent years, much attention has been dedicated to studies of how small and noisy quantum devices [1] could be used for near term applications to showcase the power of quantum computers. Besides fundamental demonstrations [2], potential applications that have been discussed are in quantum chemistry [3], discrete optimization [4] and machine learning (ML) [5–12].

Initiated by the seminal HHL algorithm [13], early work in quantum machine learning (QML) was focused on speeding up linear algebra subroutines, commonly used in ML, offering the perspective of a runtime logarithmic in the problem size [14–17]. However, most of these works have an inverse polynomial scaling of the runtime in the error and it was shown rigorously by Ciliberto et al. [18] that due to the quantum mechanical measurement process a runtime complexity $O(\sqrt{n})$ is necessary for convergence rate $1/\sqrt{n}$.

Rather than speeding up linear algebra subroutines, we focus on more recent suggestions that use a quantum device to define and implement the function class and do the optimization on a classical computer. There are two ways to that: the first are so-called *Quantum Neural Networks* (QNN) or

---

[*]JMK and SB contributed equally and are ordered randomly.

35th Conference on Neural Information Processing Systems (NeurIPS 2021).

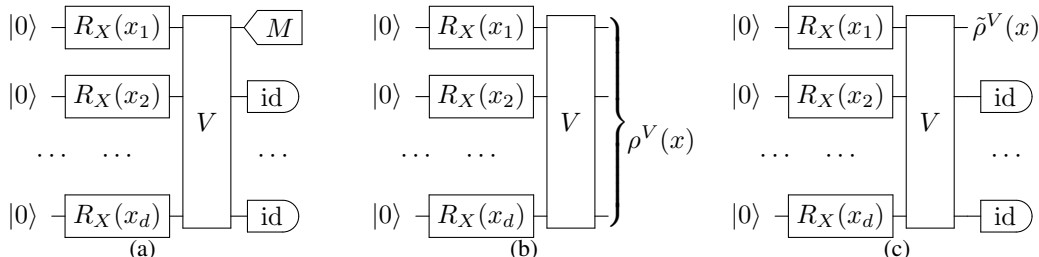

Figure 1: **Quantum advantage via inductive bias:** (a) Data generating quantum circuit $f(x) = \text{Tr}\left[\rho^V(x)(M \otimes \text{id})\right] = \text{Tr}\left[\tilde{\rho}^V(x)M\right]$. (b) The full quantum kernel $k(x,x') = \text{Tr}\left[\rho^V(x)\rho^V(x')\right]$ is too general and cannot learn $f$ efficiently. (c) The biased quantum kernel $q(x,x') = \text{Tr}\left[\tilde{\rho}^V(x)\tilde{\rho}^V(x')\right]$ meaningfully constrains the function space and allows to learn $f$ with little data.

parametrized quantum circuits [5–7] which can be trained via gradient based optimization [5, 19–23]. The second approach is to use a predefined way of encoding the data in the quantum system and defining a *quantum kernel* as the inner product of two quantum states [7–11]. These two approaches essentially provide a parametric and a non-parametric path to quantum machine learning, which are closely related to each other [11]. Since the optimization of QNNs is non-convex and suffers from so-called Barren Plateaus [24], we here focus on quantum kernels, which allow for convex problems and thus lend themselves more readily to theoretical analysis.

The central idea of using a QML model is that it enables to do computations that are exponentially hard classically. However, also in classical ML, kernel methods allow us to implicitly work with high- or infinite dimensional function spaces [25, 26]. Thus, purely studying the expressivity of QML models [27] is not sufficient to understand when we can expect speed-ups. Only recently first steps where taken into this direction [10, 12, 28]. Assuming classical hardness of computing discrete logarithms, Liu et al. [10] proposed a task based on the computation of the discrete logarithm where the quantum computer, equipped with the right feature mapping, can learn the target function with exponentially less data than any classical (efficient) algorithm. Similarly, Huang et al. [12] analyzed generalization bounds and realized that the expressivity of quantum models can hinder generalization. They proposed a heuristic to optimize the labels of a dataset such that it can be learned well by a quantum computer but not a classical machine.

In this work, we relate the discussion of quantum advantages to the classical concept of *inductive bias*. The *no free lunch* theorem informally states that no learning algorithm can outperform other algorithms on all problems. This implies that an algorithm that performs well on one type of problem necessarily performs poorly on other problems. A standard inductive bias in ML is to prefer functions that are continuous. An algorithm with that bias, however, will then struggle to learn functions that are discontinuous. For a QML model to have an edge over classical ML models, we could thus ensure that it is equipped with an inductive bias that cannot be encoded (efficiently) with a classical machine. If a given dataset fits this inductive bias, the QML model will outperform any classical algorithm. For kernel methods, the qualitative concept of inductive bias can be formalized by analyzing the spectrum of the kernel and relating it to the target function [25, 29–33].

Our main contribution is the analysis of the inductive bias of quantum machine learning models based on the spectral properties of quantum kernels. First, we show that quantum kernel methods will fail to generalize as soon as the data embedding into the quantum Hilbert space is too expressive (Theorem 1). Then we note that projecting the quantum kernel appropriately allows to construct inductive biases that are hard to create classically (Figure 1). However, our Theorem 2 also implies that estimating the biased kernel requires exponential measurements, a phenomenon reminiscent of the Barren plateaus observed in quantum neural networks. Finally we show experiments supporting our main claims.

While our work gives guidance to find a quantum advantage in ML, this yields no recipe for obtaining a quantum advantage on a classical dataset. We conjecture that unless we have a clear idea how the data generating process can be described with a quantum computer, we cannot expect an advantage by using a quantum model in place of a classical machine learning model.

## 2 Supervised learning

We briefly introduce the setting and notation for supervised learning as a preparation for our analysis of quantum mechanical methods in this context. The goal of supervised learning is the estimation of a functional mechanism based on data generated from this mechanism. For concreteness we focus on the regression setting where we assume data is generated according to $Y = f^*(X) + \varepsilon$ where $\varepsilon$ denotes zero-mean noise. We focus on $X \in \mathcal{X} \subset \mathbb{R}^d$, $Y \in \mathbb{R}$. We denote the joint probability distribution of $(X, Y)$ by $\mathcal{D}$ and we are given $n$ i.i.d. observations $D_n$ from $\mathcal{D}$. We will refer to the marginal distribution of $X$ as $\mu$, define the $L^2_\mu$ inner product $\langle f, g \rangle = \int f(x)g(x)\, \mu(\mathrm{d}x)$ and denote the corresponding norm by $\| \cdot \|$. The least square risk and the empirical risk of some hypothesis $h : \mathcal{X} \to \mathbb{R}$ is defined by $R(h) = \mathbb{E}_{\mathcal{D}} \left[ (h(X) - Y)^2 \right]$ and $R_n(h) = \mathbb{E}_{D_n} \left[ (h(X) - Y)^2 \right]$.

In supervised machine learning, one typically considers a hypothesis space $H$ of functions $h : \mathcal{X} \to \mathbb{R}$ and tries to infer $\mathrm{argmin}_{h \in H} R(h)$ (assuming for simplicity that the minimizer exists). Typically this is done by (regularized) empirical risk minimization $\mathrm{argmin}_{h \in H} R_n(h) + \lambda \Omega(h)$, where $\lambda > 0$ and $\Omega$ determine the regularization. The risk of $h$ can then be decomposed in generalization and training error $R(h) = (R(h) - R_n(h)) + R_n(h)$.

**Kernel ridge regression.** We will focus on solving the regression problem over a reproducing kernel Hilbert space (RKHS) [25, 26]. An RKHS $\mathcal{F}$ associated with a positive definite kernel $k : \mathcal{X} \times \mathcal{X} \to \mathbb{R}$ is the space of functions such that for all $x \in \mathcal{X}$ and $h \in \mathcal{F}$ the *reproducing* property $h(x) = \langle h, k(x, \cdot) \rangle_{\mathcal{F}}$ holds. Kernel ridge regression regularizes the RKHS norm, i.e., $\Omega(h) = \|h\|^2_{\mathcal{F}}$. With observations $\{(x^{(i)}, y^{(i)})\}_{i=1}^n$ we can compute the kernel matrix $K(X, X)_{ij} = k(x^{(i)}, x^{(j)})$ and the Representer Theorem [34] ensures that the empirical risk minimizer of kernel ridge regression is of the form $\hat{f}_n^\lambda(\cdot) = \sum_{i=1}^n \alpha_i k(x^{(i)}, \cdot)$, with $\alpha = (K(X, X) + \lambda\, \mathrm{id})^{-1} y$. The goal of our work is to study when a (quantum) kernel is suitable for learning a particular problem. The central object to study this is the integral operator.

**Spectral properties and inductive bias.** For kernel $k$ and marginal distribution $\mu$, the integral operator $K$, is defined as $(Kf)(x) = \int k(x, x') f(x') \mu(\mathrm{d}x')$. Mercer's Theorem ensures that there exist a spectral decomposition of $K$ with (possibly infinitely many) eigenvalues $\gamma_i$ (ordered non-increasingly) and corresponding eigenfunctions $\phi_i$, which are orthonormal in $L^2_\mu$, i.e., $\langle \phi_i, \phi_j \rangle = \delta_{i,j}$. We will assume that $\mathrm{Tr}[K] = \sum_i \gamma_i = 1$ which we can ensure by rescaling the kernel. We can then write $k(x, x') = \sum_i \gamma_i \phi_i(x) \phi_i(x')$. While the functions $\phi$ form a basis of $\mathcal{F}$ they might not completely span $L^2_\mu$. In this case we simply complete the basis and implicitly take $\gamma = 0$ for the added functions. Then we can decompose functions in $L^2_\mu$ as

$$f(x) = \sum_i a_i \phi_i(x). \tag{1}$$

We have $\|f\|^2 = \sum_i a_i^2$ and $\|f\|^2_{\mathcal{F}} = \sum_i \frac{a_i^2}{\gamma_i}$ (if $f \in \mathcal{F}$). Kernel ridge regression penalizes the RKHS norm of functions. The components corresponding to zero eigenvalues are infinitely penalized and cannot be learned since they are not in the RKHS. For large regularization $\lambda$ the solution $\hat{f}_n^\lambda$ is heavily *biased* towards learning only the coefficients of the principal components (corresponding to the largest eigenvalues) and keeps the other coefficients small (at the risk of *underfitting*). Decreasing the regularization allows ridge regression to also fit the other components, however, at the potential risk of overfitting to the noise in the empirical data. Finding good choices of $\lambda$ thus balances this *bias-variance* tradeoff.

We are less concerned with the choice of $\lambda$, but rather with the spectral properties of a kernel that allow for a quantum advantage. Similar to the above considerations, a target function $f$ can easily be learned if it is well *aligned* with the principal components of a kernel. In the easiest case, the kernel only has a single non-zero eigenvalue and is just $k(x, x') = f(x)f(x')$. Such a construction is arguably the simplest path to a quantum advantage in ML.

**Example 1** (Trivial Quantum Advantage). Let $f$ be a scalar function that is easily computable on a quantum device but requires exponential resources to approximate classically. Generate data as $Y = f(X) + \epsilon$. The kernel $k(x, x') = f(x)f(x')$ then has an exponential advantage for learning $f$ from data.

To go beyond this trivial case, we introduce two qualitative measures to judge the quality of a kernel for learning the function $f$. The *kernel target alignment* of Cristianini et al. [30] is

$$A(k, f) = \frac{\langle k, f \otimes f \rangle}{\langle k, k \rangle^{1/2} \langle f \otimes f, f \otimes f \rangle^{1/2}} = \frac{\sum_i \gamma_i a_i^2}{(\sum_i \gamma_i^2)^{1/2} \sum_i a_i^2} \tag{2}$$

and measures how well the kernel fits $f$. If $A = 1$, learning reduces to estimating a single real parameter, whereas for $A = 0$, learning is infeasible. We note that the kernel target alignment also weighs the contributions of $f$ depending on the corresponding eigenvalue, i.e., the alignment is better if large $|a_i|$ correspond to large $\gamma_i$. The kernel target alignment was used extensively to optimize kernel functions [31] and recently also used to optimize quantum kernels [35].

In a similar spirit, the *task-model alignment* of Canatar et al. [32] measures how much of the signal of $f$ is captured in the first $i$ principal components: $C(i) = \sum_{j \leq i} a_j^2 (\sum_j a_j^2)^{-1}$. The slower $C(i)$ approaches 1, the harder it is to learn as the target function is more spread over the eigenfunctions.

## 3 Quantum computation in machine learning

In this section we introduce hypothesis spaces containing functions whose output is given by the result of a quantum computation. For a general introduction to concepts of quantum computation we refer to the book of Nielsen and Chuang [36].

We will consider quantum systems comprising $d \in \mathbb{N}$ qubits. Discussing such systems and their algebraic properties does not require in-depth knowledge of quantum mechanics. A *pure state* of a single qubit is described by vector $(\alpha, \beta)^\top \in \mathbb{C}^2$ s.t. $|\alpha|^2 + |\beta|^2 = 1$ and we write $|\psi\rangle = \alpha |0\rangle + \beta |1\rangle$, where $\{|0\rangle, |1\rangle\}$ forms the computational basis. A $d$ qubit pure state lives in the tensor product of the single qubit state spaces, i.e., it is described by a normalized vector in $\mathbb{C}^{2^d}$. A *mixed state* of a $d$-qubit system can be described by a density operator $\rho \in \mathbb{C}^{2^d \times 2^d}$, i.e., a positive definite matrix ($\rho \geq 0$) with unit trace ($\mathrm{Tr}[\rho] = 1$). For a pure state $|\psi\rangle$ the corresponding density operator is $\rho = |\psi\rangle \langle\psi|$ (here, $\langle\psi|$ is the complex conjugate transpose of $|\psi\rangle$). A general density operator can be thought of as a classical probabilistic mixture of pure states. We can extract information from $\rho$ by estimating (through repeated measurements) the expectation of a suitable *observable*, i.e., a Hermitian operator $M = M^\dagger$ (where the adjoint $(\cdot)^\dagger$ is the complex conjugate of the transpose), as

$$\mathrm{Tr}[\rho M]. \tag{3}$$

Put simply, the potential advantage of a quantum computer arises from its state space being exponentially large in the number of qubits $d$, thus computing general expressions like (3) on a classical computer is exponentially hard. However, besides the huge obstacles in building quantum devices with high fidelity, the fact that the outcome of the quantum computation (3) has to be estimated from measurements often prohibits to easily harness this power, see also Wang et al. [37], Peters et al. [38]. We will discuss this in the context of quantum kernels in Section 4.

We consider parameter dependent quantum states $\rho(x) = U(x)\rho_0 U^\dagger(x)$ that are generated by evolving the initial state $\rho_0$ with the data dependent unitary transformation $U(x)$ [7, 11]. Most often we will without loss of generality assume that the initial state is $\rho_0 = (|0\rangle \langle 0|)^{\otimes d}$. We then define quantum machine learning models via observables $M$ of the data dependent state

$$f_M(x) = \mathrm{Tr}\left[U(x)\rho_0 U^\dagger(x)M\right] = \mathrm{Tr}[\rho(x)M]. \tag{4}$$

In the following we introduce the two most common function classes suggested for quantum machine learning. We note that there also exist proposals that do not fit into the form of Eq. (4) [27, 35, 39].

**Quantum neural networks.** A "quantum neural network" (QNN) is defined via a *variational quantum circuit* (VQC) [6, 40, 41]. Here the observable $M_\theta$ is parametrized by $p \in \mathbb{N}$ classical parameters $\theta \in \Theta \subseteq \mathbb{R}^p$. This defines a parametric function class $\mathcal{F}_\Theta = \{f_{M_\theta} | \theta \in \Theta\}$. The most common ansatz is to consider $M_\theta = U(\Theta)MU^\dagger(\Theta)$ where $U(\Theta) = \prod_i U(\theta_i)$ is the composition of unitary evolutions each acting on few qubits. For this and other common models of the parametric circuit it is possible to analytically compute gradients and specific optimizers for quantum circuits based on gradient descent have been developed [5, 19–23]. Nevertheless, the optimization is usually a non-convex problem and suffers from additional difficulties due to oftentimes exponentially (in $d$)

Table 1: Concepts in the quantum Hilbert space $\mathcal{H}$ and the reproducing kernel Hilbert space $\mathcal{F}$.

| Quantum Space of $d$ qubits | RKHS |
|---|---|
| $x \mapsto \rho(x) \in \mathcal{H}$ (explicit feature map) $\quad$ $\mathcal{H} = \left\{ \rho \in \mathbb{C}^{2^d \times 2^d} \mid \rho = \rho^\dagger, \rho \geq 0, \operatorname{Tr}[\rho] = 1 \right\}$ | $x \mapsto k(\cdot, x) \in \mathcal{F}$ (canonical feature map) |
| $k(x, x') = \operatorname{Tr}[\rho(x)\rho(x')] = \langle \rho(x), \rho(x') \rangle_{\mathcal{H}}$ | $k(x, x') = \langle k(\cdot, x), k(\cdot, x') \rangle_{\mathcal{F}}$ |
| $\mathcal{F} = \{ f_M \mid f_M(\cdot) = \operatorname{Tr}[\rho(\cdot)M], M = M^\dagger \}$ | $\mathcal{F} = \overline{\operatorname{Span}}\left( \{ k(\cdot, x) \mid x \in \mathcal{X} \} \right)$ |

vanishing gradients [24]. This hinders a theoretical analysis. Note that the non-convexity does not arise from the fact that the QNN can learn non-linear functions, but rather because the observable $M_\theta$ depends non-linearly on the parameters. In fact, the QNN functions are linear in the *fixed* feature mapping $\rho(x)$. Therefore the analogy to classical neural networks is somewhat incomplete.

**Quantum kernels.** The class of functions we consider are RKHS functions where the kernel is expressed by a quantum computation. The key observation is that (4) is linear in $\rho(x)$. Instead of optimizing over the parametric function class $\mathcal{F}_\Theta$, we can define the nonparametric class of functions $\mathcal{F} = \{ f_M \mid f_M(\cdot) = \operatorname{Tr}[\rho(\cdot)M], M = M^\dagger \}$.[2] To endow this function class with the structure of an RKHS, observe that the expression $\operatorname{Tr}[\rho_1 \rho_2]$ defines a scalar product on density matrices. We then define kernels via the inner product of data-dependent density matrices:

**Definition 1** (Quantum Kernel [7, 8, 11]). Let $\rho : x \mapsto \rho(x)$ be a fixed feature mapping from $\mathcal{X}$ to density matrices. Then the corresponding *quantum kernel* is $k(x, x') = \operatorname{Tr}[\rho(x)\rho(x')]$.

The Representer Theorem [34] reduces the empirical risk minimization over the exponentially large function class $\mathcal{F}$ to an optimization problem with a set of parameters whose dimensionality corresponds to the training set size. Since the ridge regression objective is convex (and so are many other common objective functions in ML), this can be solved efficiently with a classical computer.

In the described setting, the quantum computer is only used to estimate the kernel. For pure state encodings, this is done by inverting the data encoding transformation (taking its conjugate transpose) and measuring the probability that the resulting state equals the initial state $\rho_0$. To see this we use the cyclic property of the trace $k(x, x') = \operatorname{Tr}[\rho(x)\rho(x')] = \operatorname{Tr}[U(x)\rho_0 U^\dagger(x) U(x')\rho_0 U^\dagger(x')] = \operatorname{Tr}[(U^\dagger(x')U(x)\rho_0 U^\dagger(x)U(x'))\, \rho_0]$. If $\rho_0 = (|0\rangle \langle 0|)^{\otimes d}$, then $k(x, x')$ corresponds to the probability of observing every qubit in the '0' state after the initial state was evolved with $U^\dagger(x')U(x)$. To evaluate the kernel, we thus need to estimate this probability from a finite number of measurements. For our theoretical analysis we work with the exact value of the kernel and in our experiments we also simulate the full quantum state. We discuss the difficulties related to measurements in Sec. 4.

## 4 The inductive bias of simple quantum kernels

We now study the inductive bias for simple quantum kernels and their learning performance. We first give a high level discussion of a general hurdle for quantum machine learning models to surpass classical methods and then analyze two specific kernel approaches in more detail.

**Continuity in classical machine learning.** Arguably the most important bias in nonparametric regression are continuity assumptions on the regression function. This becomes particularly apparent in, e.g., nearest neighbour regression or random regression forests [42] where the regression function is a weighted average of close points. Here we want to emphasize that there is a long list of results concerning the minimax optimality of kernel methods for regression problems [43–45]. In particular these results show that asymptotically the convergence of kernel ridge regression of, e.g., Sobolev functions reaches the statistical limits which also apply to any quantum method.

---

[2] $\mathcal{F}$ is defined for a fixed feature mapping $x \mapsto \rho(x)$. Although $M$ is finite dimensional and thus $\mathcal{F}$ can be seen as a parametric function class, we will be interested in the case where $M$ is exponentially large in $d$ and we can only access functions from $\mathcal{F}$ implicitly. Therefore we refer to it as nonparametric class of functions.

**A simple quantum kernel.** We now restrict our attention to rather simple kernels to facilitate a theoretical analysis. As indicated above we consider data in $\mathcal{X} \subset \mathbb{R}^d$ and we assume that the distribution $\mu$ of the data factorizes over the coordinates (i.e. $\mu$ can be written as $\mu = \bigotimes \mu_i$). This data is embedded in a $d$-qubit quantum circuit. Let us emphasize here that the RKHS based on a quantum state of $d$-qubits is at most $4^d$ dimensional, i.e., finite dimensional and in the infinite data limit $n \to \infty$ standard convergence guarantees from parametric statistics apply. Here we consider growing dimension $d \to \infty$, and sample size polynomial in the dimension $n = n(d) \in \mathrm{Poly}(d)$. In particular the sample size $n \ll 4^d$ will be much smaller than the dimension of the feature space and bounds from the parametric literature do not apply.

Here we consider embeddings where each coordinate is embedded into a single qubit using a map $\varphi_i$ followed by an arbitrary unitary transformation $V$, so that we can express the embedding in the quantum Hilbert space as $|\psi^V(x)\rangle = V \bigotimes |\varphi_i(x_i)\rangle$ with corresponding density matrix (feature map)[3]

$$\rho^V(x) = |\psi^V(x)\rangle \langle \psi^V(x)|. \tag{5}$$

Note that the corresponding kernel $k(x, x') = \mathrm{Tr}\left[\rho(x)\rho(x')\right]$ is independent of $V$ and factorizes $k(x, x') = \mathrm{Tr}\left[\bigotimes \rho_i(x_i) \bigotimes \rho_i(x'_i)\right] = \prod \mathrm{Tr}\left[\rho_i(x_i)\rho_i(x'_i)\right]$ where $\rho_i(x_i) = |\varphi_i(x_i)\rangle\langle\varphi_i(x_i)|$. The product structure of the kernel allows us to characterize the RKHS generated by $k$ based on the one dimensional case. The embedding of a single variable can be parametrized by complex valued functions $a(x), b(x)$ as

$$|\varphi_i(x)\rangle = a(x)|0\rangle + b(x)|1\rangle. \tag{6}$$

One important object characterizing this embedding turns out to be the mean density matrix of this embedding given by $\rho_{\mu_i} = \int \rho_i(y)\,\mu_i(\mathrm{d}y) = \int |\varphi_i(y)\rangle\langle\varphi_i(y)|\,\mu_i(\mathrm{d}y)$. This can be identified with the kernel mean embedding of the distribution [46]. Note that for factorizing base measure $\mu$ the factorization $\rho_\mu = \bigotimes \rho_{\mu_i}$ holds. Let us give a concrete example to clarify the setting, see Fig. 1(b).

**Example 2.** [11, Example III.1.] We consider the cosine kernel where $a(x) = \cos(x/2)$, $b(x) = i\sin(x/2)$. This embedding can be realized using a single quantum $R_X(x) = \exp\left(-i\frac{x}{2}\sigma_x\right)$ gate such that $|\psi(x)\rangle = R_X(x)|0\rangle = \cos(x/2)|0\rangle + i\sin(x/2)|1\rangle$. In this case the kernel is given by

$$k(x, x') = |\langle 0|R_X^\dagger(x)R_X(x)|0\rangle|^2 = |\cos(\tfrac{x}{2})\cos(\tfrac{x'}{2}) + \sin(\tfrac{x}{2})\sin(\tfrac{x'}{2})|^2 = \cos(\tfrac{x-x'}{2})^2. \tag{7}$$

As a reference measure $\mu$ we consider the uniform measure on $[-\pi, \pi]$. Then the mean density matrix is the completely mixed state $\rho_\mu = \frac{1}{2}\,\mathrm{id}$. For $\mathbb{R}^d$ valued data whose coordinates are encoded independently the kernel is given by $k(x, x') = \prod \cos^2\left((x_i - x'_i)/2\right)$ and $\rho_\mu = 2^{-d}\mathrm{id}_{2^d \times 2^d}$. We emphasize that due to the kernel trick this kernel can be evaluated classically in runtime $O(d)$.

**Quantum RKHS.** We now characterize the RKHS and the eigenvalues of the integral operator for quantum kernels. The RKHS consists of all functions $f \in \mathcal{F}$ that can be written as $f(x) = \mathrm{Tr}\left[\rho(x)M\right]$ where $M \in \mathbb{C}^{2^d \times 2^d}$ is a Hermitian operator. Using this characterization of the finite dimensional RKHS we can rewrite the infinite dimensional eigenvalue problem of the integral operator as a finite dimensional problem. The action of the corresponding integral operator on $f$ can be written as

$$(Kf)(x) = \int f(y)k(y, x)\,\mu(\mathrm{d}y) = \int \mathrm{Tr}\left[M\rho(y)\right]\mathrm{Tr}\left[\rho(y)\rho(x)\right]\mu(\mathrm{d}y)$$
$$= \int \mathrm{Tr}\left[(M \otimes \rho(x))(\rho(y) \otimes \rho(y))\right]\mu(\mathrm{d}y) = \mathrm{Tr}\left[(M \otimes \rho(x))\int \rho(y) \otimes \rho(y)\,\mu(\mathrm{d}y)\right] \tag{8}$$

We denote the operator $O_\mu = \int \rho(y) \otimes \rho(y)\,\mu(\mathrm{d}y)$ for which $\mathrm{Tr}\left[O_\mu\right] = 1$ holds. Then we can write

$$(Kf)(x) = \mathrm{Tr}\left[O_\mu(M \otimes \rho(x))\right] = \mathrm{Tr}\left[O_\mu(M \otimes \mathrm{id})(\mathrm{id} \otimes \rho(x))\right]$$
$$= \mathrm{Tr}\left[\mathrm{Tr}_1\left[O_\mu(M \otimes \mathrm{id})\right]\rho(x)\right] \tag{9}$$

---

[3] When we can ignore $V$, we simply assume $V = \mathrm{id}$ and write $\rho(x)$ instead of $\rho^V(x)$. For the kernel, since $V^\dagger V = \mathrm{id} = V^\dagger V$ and due to the cyclic property of the trace we have $k^V(x, x') = \mathrm{Tr}\left[\rho^V(x)\rho^V(x')\right] = \mathrm{Tr}\left[V\rho(x)V^\dagger V\rho(x')V^\dagger\right] = \mathrm{Tr}\left[V^\dagger V\rho(x)V^\dagger V\rho(x')\right] = \mathrm{Tr}\left[\rho(x)\rho(x')\right] = k(x, x')$.

where $Tr_1 [\cdot]$ refers to the partial trace over the first factor. For the definition and a proof of the last equality we refer to Appendix A. The eigenvalues of $K$ can now be identified with the eigenvalues of the linear map $T_\mu$ mapping $M \rightarrow Tr_1 [O_\mu(M \otimes \mathrm{id})]$. As shown in the appendix there is an eigendecomposition such that $T_\mu(M) = \sum \lambda_i A_i Tr [A_i M]$ where $A_i$ are orthonormal Hermitian matrices (for details, a proof and an example we refer to Appendix C). The eigenfunctions of $K$ are given by $f_i(x) = Tr [\rho(x)A_i]$.

We now state a bound that controls the largest eigenvalue of the integral operator $K$ in terms of the eigenvalues of the mean density matrix $\rho_\mu$ (Proof in Appendix C.2).

**Lemma 1.** *The largest eigenvalue $\gamma_{max}$ of $K$ satisfies the bound $\gamma_{max} \leq \sqrt{Tr \left[\rho_\mu^2\right]}$.*

The lemma above shows that the squared eigenvalues of $K$ are bounded by $Tr \left[\rho_\mu^2\right]$, an expression known as the *purity* [36] of the density matrix $\rho_\mu$, which measures the diversity of the data embedding. On the other hand the eigenvalues of $K$ are closely related to the learning guarantees of kernel ridge regression. In particular, standard generalization bounds for kernel ridge regression [47] become vacuous when $\gamma_{max}$ is exponentially smaller than the training sample size (if $Tr [K] = 1$ which holds for pure state embeddings). The next result shows that this is not just a matter of bounds.

**Theorem 1.** *Suppose the purity of the embeddings $\rho_{\mu_i}$ satisfies $Tr \left[\rho_{\mu_i}^2\right] \leq \delta < 1$ as the dimension and number of qubits $d$ grows. Furthermore, suppose the training sample size only grows polynomially in $d$, i.e., $n \leq d^l$ for some fixed $l \in \mathbb{N}$. Then there exists $d_0 = d_0(\delta, l, \varepsilon)$ such that for all $d \geq d_0$ no function can be learned using kernel ridge regression with the $d$-qubit kernel $k(x, x') = Tr [\rho(x)\rho(x')]$ in the sense that for any $f \in L^2$, with probability at least $1 - \varepsilon$ for all $\lambda \geq 0$*

$$R(\hat{f}_n^\lambda) \geq (1 - \varepsilon)\|f\|^2. \tag{10}$$

The proof of the theorem can be found in Appendix D. It relies on a general result (Theorem 3 in Appendix D) which shows that for any (not necessarily quantum) kernel the solution of kernel ridge regression cannot generalize when the largest eigenvalue in the Mercer decomposition is sufficiently small (depending on the sample size). Then the proof of Theorem 1 essentially boils down to proving a bound on the largest eigenvalue using Lemma 1.

Theorem 1 implies that generalization is only possible when the mean embedding of most coordinates is close to a pure state, i.e. the embedding $x \rightarrow |\varphi_i(x)\rangle$ is almost constant. To make learning from data feasible we cannot use the full expressive power of the quantum Hilbert space but instead only very restricted embeddings allow to learn from data. This generalizes an observation already made in [12]. Since also classical methods allow to handle high-dimensional and infinite dimensional RKHS the same problem occurs for classical kernels where one solution is to adapt the bandwidth of the kernel to control the expressivity of the RKHS. In principle this is also possible in the quantum context, e.g., for the cosine embedding.

**Biased kernels.** We have discussed that without any inductive bias, the introduced quantum kernel cannot learn any function for large $d$. One suggestion to reduce the expressive power of the kernel is the use of *projected* kernels [12]. They are defined using reduced density matrices given by $\tilde{\rho}_m^V(x) = Tr_{m+1...d} \left[\rho^V(x)\right]$ where $Tr_{m+1...d} [\cdot]$ denotes the partial trace over qubits $m + 1$ to $d$ (definition in Appendix A) . Then they consider the usual quantum kernel for this embedding $q_m^V(x, x') = Tr \left[\tilde{\rho}_m^V(x)\tilde{\rho}_m^V(x')\right]$. Physically, this corresponds to just measuring the first $m$ qubits and the functions $f$ in the RKHS can be written in terms of a hermitian operator $M$ acting on $m$ qubits so that $f(x) = Tr \left[\rho^V(x)(M \otimes \mathrm{id})\right] = Tr \left[\tilde{\rho}_m^V(x)M\right]$. If $V$ is sufficiently complex it is assumed that $f$ is hard to compute classically [48].

Indeed above procedure reduces the generalization gap. But this comes at the price of an increased *approximation error* if the remaining RKHS cannot fully express the target function $f^*$ anymore, i.e., the learned function *underfits*. Without any reason to believe that the target function is well represented via the projected kernel, we cannot hope for a performance improvement by simply reducing the size of the RKHS in an arbitrary way. However, if we *know* something about the data generating process than this can lead to a meaningful inductive bias. For the projected kernel this could be that we *know* that the target function can be expressed as $f^*(x) = Tr \left[\tilde{\rho}_m^V(x)M^*\right]$, see Fig. 1. In this case using $q_m^V$ improves the generalization error without increasing the approximation error. To emphasize this, we will henceforth refer to $q_m^V$ as *biased kernel*.

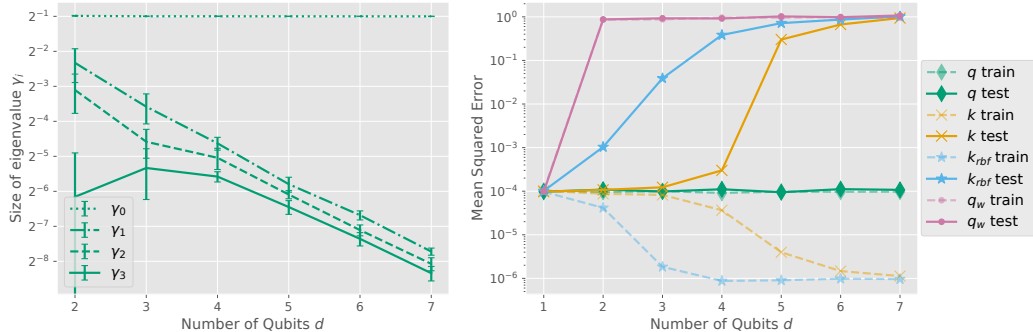

Figure 2: **Left:** Spectral behavior of biased kernel $q$, see Theorem 2b) and Equation (11) **Right:** The biased kernel $q$, equipped with prior knowledge, easily learns the function for arbitrary number of qubits and achieves optimal mean squared error (MSE). Models that are ignorant to the structure of $f^*$ fail to learn the function. The classical kernel $k_{rbf}$ and the full quantum kernel overfit (they have low training error, but large test error). The biased kernel on the wrong qubit $q_w$ has litle capacity with the wrong bias and thus underfits (training and test error essentially overlap).

We now investigate the RKHS for reduced density matrices where $V$ is a Haar-distributed random unitary matrix (proof in Appendix E).

**Theorem 2.** *Suppose $V$ is distributed according to the Haar measure on the group of unitary matrices. Fix $m$. Then the following two statements hold:*

*a) The reduced density operator satisfies with high probability $\tilde{\rho}_m^V = 2^{-m}\mathrm{id} + O(2^{-d/2})$ and the projected kernel satisfies with high probability $q_m^V(x, x') = 2^{-m} + O(2^{-d/2})$ as $d \to \infty$.*

*b) Let $T_{\mu,m}^V$ denote the linear integral operator for the kernel $q_m^V$ as defined above. Then the averaged operator $\mathbb{E}_V\left[T_{\mu,m}^V\right]$ has one eigenvalue $2^{-m} + O(2^{-2d})$ whose eigenfunction is constant (up to higher order terms of order $O(2^{-2d})$ and $2^{2m} - 1$ eigenvalues $2^{-m-d} + O(2^{-2d})$.*

The averaged integral operator in the second part of the result is not directly meaningful, however it gives some indication of the behavior of the operators $T_{\mu,m}^V$. In particular, we expect a similar result to hold for most $V$ if the mean embedding $\rho_\mu$ is sufficiently mixed. A proof of this result would require us to bound the variance of the matrix elements of $T_{\mu,m}^V$ which is possible using standard formula for expectations of polynomials over the unitary group but lengthy.

Note that the dimension of the RKHS for the biased kernel $q_m^V$ with $m$-qubits is bounded by $4^m$. This implies that learning is possible when projecting to sufficiently low dimensional biased kernels such that the training sample size satisfies $n \gtrsim 4^m \geq \dim(\mathcal{F})$.

Let us now focus on the case $m = 1$, that is the biased kernel is solely defined via the first qubit. Assuming that Theorem 2b) also holds for fixed $V$ we can assume that the biased kernel has the form

$$q(x, x') \equiv q_1^V(x, x') = \gamma_0\phi_0(x)\phi_0(x') + \sum\nolimits_{i=1}^{3} \gamma_i\phi_i(x)\phi_i(x'), \tag{11}$$

where $\gamma_0 = 1/2 + O(2^{-2d})$ and $\phi_0(x) = 1$ is the constant function up to terms of order $O(2^{-2d})$. For $i = 1, 2, 3$ we have $\gamma_i = O(2^{-d-1}) = O(2^{-d})$ (Fig. 2) and $\phi_i$ is a function that conjectured to be exponentially hard in $d$ to compute classically [48]. It is thus impossible to include a bias towards those three eigenfunctions classically. On the other hand we can include a strong bias towards the constant eigenfunction also classically. The straightforward way to do this is to center the data in the RKHS (see Appendix B for details).

**Barren plateaus.** Another conclusion from Theorem 2a) is that the fluctuations of the reduced density matrix around its mean are exponentially vanishing in the number of qubits. In practice the value of the kernel would be determined by measurements and exponentially many measurements are necessary to obtain exponential accuracy of the kernel function. Therefore the theorem suggests that it is not possible in practice to learn anything beyond the constant function from generic biased

kernels for (modestly) large values of $d$. This observation is closely related to the fact that for many quantum neural networks architectures, the gradient of the parameters with respect to the loss is exponentially vanishing with the system size $d$, a phenomenon known as *Barren plateaus* [24, 49].

## 5    Experiments

Since for small $d$ we can simulate the biased kernel efficiently, we illustrate our theoretical findings in the following experiments. Our implementation, building on standard open source packages [50, 51], is available online.[4] We consider the case described above where we *know* that the data was generated by measuring an observable on the first qubit, i.e., $f^*(x) = \text{Tr}\left[\tilde{\rho}_1^V(x)M^*\right]$, but we do not know $M^*$, see Fig. 1. We use the full kernel $k$ and the biased kernel $q$ for the case $m = 1$. To show the effect of selecting the wrong bias, we also include the behavior of a biased kernel defined only on the second qubit, denoted as $q_w$. As a classical reference we also include the performance of a radial basis function kernel $k_{\text{rbf}}(x, x') = \exp(-\|x - x'\|^2/2)$. For the experiments we fix a single qubit observable $M^* = \sigma_z$ and perform the experiment for varying number $d$ of qubits. First we draw a random unitary $V$. The dataset is then generated by drawing $N = 200$ realizations $\{x^{(i)}\}_{i=1}^N$ from the $d$ dimensional uniform distribution on $[0, 2\pi]^d$. We then define the labels as $y^{(i)} = cf^*(x^{(i)}) + \epsilon^{(i)}$, where $f^*(x) = \text{Tr}\left[\tilde{\rho}^V(x)\sigma_z\right]$, $\epsilon^{(i)}$ is Gaussian noise with $\text{Var}[\epsilon] = 10^{-4}$, and $c$ is chosen such that $\text{Var}[f(X)] = 1$. Keeping the variances fixed ensures that we can interpret the behavior for varying $d$.

We first verify our findings from Theorem 2b) and Equation (11) by estimating the spectrum of $q$. Fig. 2 (left) shows that Theorem 2b) also holds for individual $V$ with high probability. We then use $2/3$ of the data for training kernel ridge regression (we fit the mean seperately) with preset regularization, and use $1/3$ to estimate the test error. We average the results over ten random seeds (random $V$, $x^{(i)}$, $\epsilon^{(i)}$) and results are reported in the right panel of Fig. 2. This showcases that as the number of qubits increases, it is impossible to learn $f^*$ without the appropriate spectral bias. $k$ and $k_{\text{rbf}}$ have too little bias and overfit, whereas $q_w$ has the wrong bias and severly underfits. The performance of $q_w$ underlines that randomly biasing the kernel does not significantly improve the performance over the full kernel $k$. In the appendix we show that this is not due to a bad choice of regularization, by reporting cherry-picked results over a range of regularizations.

To further illustrate the spectral properties, we empirically estimate the kernel target alignment [30] and the task-model alignment [32] that we introduced in Sec. 2. By using the centered kernel matrix (see App. B) and centering the data we can ignore the first eigenvalue in (11) corresponding the constant function. In Figure 3 (left) we show the empirical (centered) kernel target alignment for 50 random seeds. The biased kernel is the only one well aligned with the task. The right panel of Fig. 3 shows the task model alignment. This shows that $f^*$ can be completely expressed with the first four components of the biased kernel, while the other kernels essentially need the entire spectrum (we use a sample size of 200, hence the empirical kernel matrix is only 200 dimensional) and thus are unable to learn. Note that the kernel $q_w$ is four dimensional, and so higher contributions correspond to functions outside its RKHS that it actually cannot even learn at all.

## 6    Discussion

We provided an analysis of the reproducing kernel Hilbert space (RKHS) and the inductive bias of quantum kernel methods. Rather than the dimensionality of the RKHS, its spectral properties determine whether learning is feasible. Working with exponentially large RKHS comes with the risk of having a correspondingly small inductive bias. This situation indeed occurs for naive quantum encodings, and hinders learning unless datasets are of exponential size. To enable learning, we necessarily need to consider models with a stronger inductive bias. Encoding a bias towards continuous functions is likely not a promising path for a quantum advantage, as this is where classical machine learning models excel.

Our results suggest that we can only achieve a quantum advantage if we *know* something about the data generating process and cannot efficiently encode this classically, yet are able use this information to bias the quantum model. We indeed observe an exponential advantage in the case where we know that the data comes from a single qubit observable and constrain the RKHS accordingly. However,

---

[4]`https://github.com/jmkuebler/quantumbias`

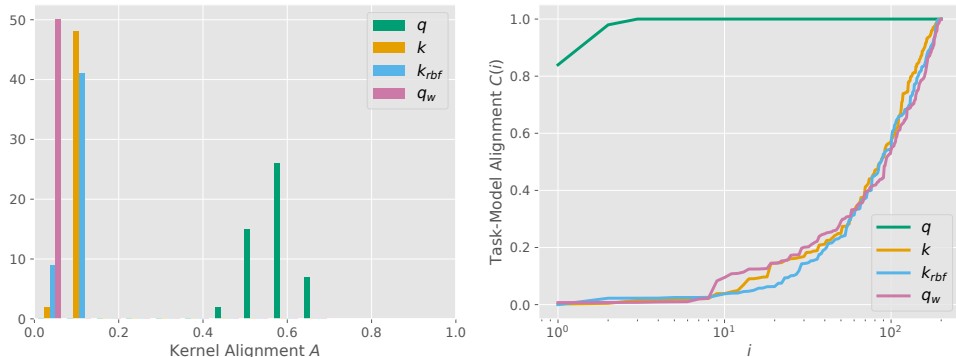

Figure 3: Histogram of the kernel target alignment over 50 runs (left) and task model alignment (right) for $d = 7$.

we find that evaluating the kernel requires exponentially many measurements, an issue related to Barren Plateaus encountered in quantum neural networks.

With fully error-corrected quantum computers it becomes feasible to define kernels with a strong bias that do not require exponentially many measurements. An example of this kind was recently presented by Liu et al. [10]: here one knows that the target function is extremely simple after computing the discrete logarithm. A quantum computer is able to encode this inductive bias by using an efficient algorithm for computing the discrete logarithm.

However, even for fully coherent quantum computers it is unclear how we can reasonably encode a strong inductive bias about a classical dataset (e.g., images of cancer cells, climate-data, etc.). The situation might be better when working with *quantum data*, i.e., data that is collected via observations of systems at a quantum mechanical scale. To summarize, we conclude that there is no indication that quantum machine learning will substantially improve supervised learning on classical datasets.

## Acknowledgments and Disclosure of Funding

The authors thank the anonymous reviewers for their helpful comments that made the theorems and their proofs more accessible. This work was in part supported by the German Federal Ministry of Education and Research (BMBF) through the Tübingen AI Center (FKZ: 01IS18039B) and the Machine Learning Cluster of Excellence, EXC number 2064/1 – Project number 390727645.

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
