# The Inductive Bias of Quantum Kernels

## Supplementary Material

## A  Partial trace in quantum mechanics

Here we provide the definition of the partial trace used for the biased quantum kernels. For details we refer to [36]. The state space of the union of two quantum systems with state space $\mathcal{H}_1$ and $\mathcal{H}_2$ is given by the tensor product $\mathcal{H}_1 \otimes \mathcal{H}_2$. A general mixed state is described by a density matrix $\rho_{12}$ which is hermitian positive linear operator on $\mathcal{H}_1 \otimes \mathcal{H}_2$ with $\mathrm{Tr}\,[\rho_{12}] = 1$. The state $\rho_1$ on the subsystem $\mathcal{H}_1$ is obtained by the partial trace operation $\rho_1 = \mathrm{Tr}_2\,[\rho_{12}]$. The partial trace can be defined as the linear map $\mathrm{Tr}_2 : \mathcal{L}(\mathcal{H}_1 \otimes \mathcal{H}_2) \to \mathcal{L}(\mathcal{H}_1)$ that satisfies

$$\mathrm{Tr}_2\,[S \otimes T] = \mathrm{Tr}\,[T]\,S \tag{12}$$

for all $S \in \mathcal{L}(\mathcal{H}_1), T \in \mathcal{L}(\mathcal{H}_2)$. It can be shown that this map exists and is unique. Picking a basis on $\mathcal{H}_1$ and $\mathcal{H}_2$ we consider the tensor product basis on $\mathcal{H}_1 \otimes \mathcal{H}_2$. In coordinates given by this basis we can write

$$(\rho_1)_{i_1 j_1} = \mathrm{Tr}_2\,[\rho_{12}]_{i_1 j_1} = \sum_{k=1}^{\dim(\mathcal{H}_2)} (\rho_{12})_{i_1 k, j_1 k}. \tag{13}$$

For completeness and to illustrate the handling of the partial trace we prove the last equality in (9). We want to show that for $S \in \mathcal{L}(\mathcal{H}_1 \otimes \mathcal{H}_2)$ and $T \in \mathcal{L}(\mathcal{H}_1)$ the identity

$$\mathrm{Tr}\,[S(T \otimes \mathrm{id}] = \mathrm{Tr}\,[\mathrm{Tr}_2\,[S]\,T] \tag{14}$$

holds. We assume first that $S = A \otimes B$ for some $A \in \mathcal{L}(\mathcal{H}_1)$ and $B \in \mathcal{L}(\mathcal{H}_2)$. Then, by definition,

$$\begin{aligned}
\mathrm{Tr}\,[\mathrm{Tr}_2\,[S]\,T] &= \mathrm{Tr}\,[\mathrm{Tr}_2\,[A \otimes B]\,T] = \mathrm{Tr}\,[AT]\,\mathrm{Tr}\,[B] \\
&= \mathrm{Tr}\,[AT \otimes B] = \mathrm{Tr}\,[(A \otimes B)(T \otimes \mathrm{id})] = \mathrm{Tr}\,[S(T \otimes \mathrm{id})].
\end{aligned} \tag{15}$$

Here we used that the trace of a tensor product is the product of the traces. For general $S$ the statement now follows from the linearity of both sides in $S$.

## B  General results about RKHS

In this section we briefly discuss basic results on centering in RKHS and the RKHS of tensor product kernels.

### B.1  Centering in the RKHS

As shown in Section 4, the constant function plays a special role for typical biased kernels as the corresponding eigenvalue is much larger than the remaining eigenvalues. Clearly, it is also possible classically to treat the constant function separately. To do so, it is natural to center the data by subtracting the mean $\bar{y} = n^{-1} \sum_{i=1}^{n} y_i$ and to consider the *centered* kernel. This corresponds to putting no penalty on the constant function which is also common in linear models where no penalty is put on the intercept. Formally, for a kernel $k$, the centered kernel is defined as

$$k_c(x, x') = k(x, x') - \mathbb{E}_X\,[k(X, x')] - \mathbb{E}_{X'}\,[k(x, X')] + \mathbb{E}_{X, X'}\,[k(X, X')]. \tag{16}$$

In analogy we can center the kernel matrix as $K_c(X, X) = \left(\mathrm{id} - \frac{1}{n}\mathbf{1}\mathbf{1}^\top\right) K(X, X) \left(\mathrm{id} - \frac{1}{n}\mathbf{1}\mathbf{1}^\top\right)$, where $\mathbf{1}$ is a vector of all ones.

Let $k$ be a kernel with Mercer decomposition

$$k(x, x') = \sum \gamma_i \phi_i(x) \phi_i(x'), \tag{17}$$

and define $a_i = \int \phi_i(x) \mu(\mathrm{d}x)$. Then the centered kernel can be written as

$$k_c(x, x') = \sum \gamma_i (\phi_i(x) - a_i)(\phi_i(x') - a_i). \tag{18}$$

To make things explicit let us focus on the biased kernel of Equation (11). Ignoring terms of order $\mathcal{O}(2^{-2d})$, the constant function is an eigenfunction of the kernel. In such a case centering corresponds to setting the corresponding eigenvalue $\gamma_0$ to zero, while the other terms in the spectral decomposition are invariant under centering (by orthogonality we have $a_i = 0$ for $i \neq 0$). The centered biased kernel of Eq. (11) is thus

$$q_c(x, x') = \sum_{i=1}^{3} \gamma_i \phi_i(x) \phi_i(x'). \tag{19}$$

By Theorem 2 we expect that all the eigenvalues of the centered biased kernel are similarly large. Further we know that the centered part of the target function can completely be expressed in terms of the eigenfunctions of the centered biased kernel $f^*(x) - \bar{f}^* = \sum_{i=1}^{3} a_i \phi_i(x)$, where $\bar{f}^* = \mathbb{E}\left[f^*(X)\right]$. Let us assume that all the three eigenvalues are completely equal. Then we can compute the kernel target alignment of Eq. (2)

$$A(q_c, f^* - \bar{f}^*) = \frac{\sum_{i=1}^{3} \gamma a_i^2}{(\sum_{i=1}^{3} \gamma^2)^{1/2} \sum_{i=1}^{3} a_i^2} = \frac{\gamma \sum_{i=1}^{3} a_i^2}{\sqrt{3}\gamma \sum_{i=1}^{3} a_i^2} = \frac{1}{\sqrt{3}} \approx 0.58. \tag{20}$$

We emphasize that this expectation is in good accordance with the results of our experiments reported in Fig. 3. Further, note that computing the kernel target alignment after centering is quite common in the kernel literature and is used to optimize the kernel function [31].

### B.2  Tensor product of kernels

In this section we describe the construction of product kernels on product spaces. More details can be found in any textbook on RKHS [25]. Let $(X_1, k_1)$ and $(X_2, k_2)$ be two spaces with positive definite kernels with RKHS $\mathcal{F}_1$ and $\mathcal{F}_2$. Then the function

$$k((x_1, x_2), (y_1, y_2)) = k_1(x_1, y_1) k_2(x_2, y_2) \tag{21}$$

defines a positive definite kernel on $X_1 \times X_2$ and the RKHS is given by $\{f_1(x_1) f_2(x_2) : f_1 \in \mathcal{F}_1, f_2 \in \mathcal{F}_2\}$. Morevoer, if we are given a product measure $\mu = \mu_1 \otimes \mu_2$ on $X_1 \times X_2$ then the integral operator for the kernel $k$ factorizes, i.e., for functions $f(x_1, x_2) = f_1(x_1) f_2(x_2)$

$$
\begin{aligned}
(Kf)(x_1, x_2) &= \int f(y) k(y, x) \, \mu(\mathrm{d}y) \\
&= \int f_1(y_1) k_1(y_1, x_1) \, \mu_1(\mathrm{d}x_1) \int f_2(y_2) k_2(y_2, x_2) \, \mu_1(\mathrm{d}x_2) \\
&= (K_1 f_1)(x_1)(K_2 f_2)(x_2).
\end{aligned} \tag{22}
$$

Therefore the eigenvalue problems of the integral operators decouple and the eigenvalues of $K$ are given by $\{\gamma^1 \gamma^2 : \gamma^1 \in E_1, \gamma^2 \in E_2\}$ where $E_i$ denotes the eigenvalues of $K_i$.

It can be derived from the results above that the RKHS of the product kernel $k(x, x') = k_1(x, x') k_2(x, x')$ on $X$ is given by $\{f_1(x) f_2(x) : f_1 \in \mathcal{F}_1, f_2 \in \mathcal{F}_2\}$ where $\mathcal{F}_i$ denotes the RKHS of $(X, k_i)$. There is no simple relation for the integral operators.

## C  More details on quantum kernels for classical data

In this section we analyze in more detail the properties of quantum kernel methods for classical data.

### C.1  Description of the RKHS

To understand the quantum kernel better we give a description of the RKHS for the quantum kernels. We consider the one-qubit embedding $x \to a(x)|0\rangle + b(x)|1\rangle$. The RKHS $\tilde{\mathcal{F}}$ corresponding to the (non-physical) kernel $\tilde{k}(x, y) = \langle \varphi(x), \varphi(y) \rangle$ is then generated by $a(x), b(x)$. Moreover, the RKHS corresponding to the physical kernel $k(x, x') = \mathrm{Tr}\left[\rho(x)\rho(x')\right] = |\langle \varphi(x), \varphi(x') \rangle|^2 = |\tilde{k}(x, x')|^2 = \tilde{k}(x, x')\overline{\tilde{k}}(x, x')$ is the vector space $\mathcal{F}$ generated by $\{f \cdot \bar{g} : f, g \in \tilde{F}\}$ [52] (to obtain the real valued RKHS which is more relevant in the learning theoretic setting we consider the real and imaginary

part). This result can also be obtained by looking at the feature map $x \to \rho(x)$ of the physical kernel directly. When we consider data from $\mathbb{R}^d$ where all dimensions are encoded independently in a single qubit the resulting RKHS has the tensor product structure $\mathcal{F} = \bigotimes \mathcal{F}_i$ where $\mathcal{F}_i$ are the RKHS for the single coordinate embeddings.

## C.2 Proof of Lemma 1

Here we analyze the integral operators in a bit more detail and prove Lemma 1. For the proof of Lemma 1 we need to briefly look again at the simpler non-physical kernel $\tilde{k}(x, y) = \langle \varphi(x), \varphi(y) \rangle$ and its integral operator. Suppose data has distribution $\mu$ on $\mathbb{R}$. We consider the integral operator $\tilde{K}$ acting on $f(x) = \langle \omega, \varphi(x) \rangle$ defined by

$$\tilde{K}f(x) = \int f(y)\tilde{k}(y, x)\, \mu(\mathrm{d}y) = \int \langle \omega, \varphi(y) \rangle \langle \varphi(y), \varphi(x) \rangle\, \mu(\mathrm{d}y) = \langle \omega, \rho_\mu \varphi(x) \rangle \quad (23)$$

where $\rho_\mu = \int |\varphi(y)\rangle\langle\varphi(y)|\, \mu(\mathrm{d}y)$ denotes the mean density matrix associated with the measure $\mu$. We observe that the eigenvalues $\tilde{\gamma}_i$ of $\tilde{K}$ agree with the the eigenvalues of the density matrix $\rho_\mu$. In particular we conclude

$$\|\tilde{K}\|_{HS}^2 = \sum \tilde{\gamma}_i^2 = \|\rho_\mu\|_{HS}^2 = \mathrm{Tr}\left[\rho_\mu^2\right] \quad (24)$$

where $\|\cdot\|_{HS}$ denotes the Hilbert-Schmidt norm (which for symmetric matrices agrees with the Frobenius norm). This observation corresponds to the fact that for the linear kernel the eigenvalues of the integral operator agree with the eigenvalues of the covariance matrix.

Now we can give the simple proof of Lemma 1. For convenience we restate the lemma.

**Lemma 2** (Lemma 1 in the main part). *The largest eigenvalue $\gamma_{max}$ of $K$ satisfies the bound* $\gamma_{max} \leq \sqrt{Tr\left[\rho_\mu^2\right]}$.

*Proof.* We observe, denoting the constant function with value 1 by $\mathbf{1}$,

$$\int \mathbf{1}(x)k(x, y)\mathbf{1}(y)\, \mu(\mathrm{d}x)\mu(\mathrm{d}y) = \int |\tilde{k}(x, y)|^2\, \mu(\mathrm{d}x)\mu(\mathrm{d}y) = \|\tilde{K}\|_{HS}^2 = \|\rho_\mu\|_{HS}^2 = \mathrm{Tr}\left[\rho_\mu^2\right] \quad (25)$$

where we used (24) in the last two steps. Suppose that $f$ is a normalized eigenfunction for the eigenvalue $\gamma_{max}$. From the Mercer decomposition we obtain

$$1 = K(x, x) \geq \gamma_{max} f(x)^2. \quad (26)$$

Hence $f$ is bounded by $\sqrt{\gamma_{max}}^{-1}$ and we conclude that

$$\gamma_{max} = \int f(x)(Kf)(x)\, \mu(\mathrm{d}x) = \int f(x)k(x, y)f(y)\, \mu(\mathrm{d}x)\mu(\mathrm{d}y)$$

$$\leq \gamma_{max}^{-1} \int \mathbf{1}(x)k(x, y)\mathbf{1}(y)\, \mu(\mathrm{d}x)\mu(\mathrm{d}y) = \gamma_{max}^{-1}\mathrm{Tr}\left[\rho_\mu^2\right]$$

where we used that $k$ is pointwise positive. This ends the proof. $\qquad \square$

Let us look at this result in our main setting where each coordinate of $d$-dimensional data is embedded in a single qubit. If the measure $\mu$ on $\mathbb{R}^d$ factorizes as $\mu = \bigotimes \mu_i$. The integral operator factorizes over the $d$ coordinates and the eigenvalues of the integral operator are given by $\{\prod_{j=1}^d \gamma_{i_j}, \gamma_{i_j} \in E_j\}$ with $E_j$ denoting the eigenvalues of the one-dimensional integral operators. In particular the largest eigenvalue will be exponentially small (in $d$) as soon as $\max(E_j) \leq \delta < 1$ for a fixed $\delta$ which holds if the individual embeddings satisfy $\mathrm{Tr}\left[\rho_{\mu_i}^2\right] < \delta$. Note that $\mathrm{Tr}\left[\rho_{\mu_i}^2\right] = 1$ if and only if the embedding is constant.

## C.3 Spectral decomposition of the integral operator

As shown in the main text the integral operator $K$ applied to $f(x) = \text{Tr}\left[\rho(x)M\right]$ can be written as

$$(Kf)(x) = \text{Tr}\left[O_\mu(M \otimes \rho(x))\right] = \text{Tr}\left[O_\mu(M \otimes \text{id})(\text{id} \otimes \rho(x))\right] = \text{Tr}\left[\text{Tr}_1\left[O_\mu(M \otimes \text{id})\right]\rho(x)\right] \tag{27}$$

where $O_\mu = \int \rho(y) \otimes \rho(y)\,\mu(\text{d}y)$. Note that this reformulation makes the isomorphism of $\mathcal{L}(\mathcal{H}, \mathcal{H}) \otimes \mathcal{L}(\mathcal{H}, \mathcal{H}) \simeq \mathcal{L}(\mathcal{H} \otimes \mathcal{H}, \mathcal{H} \otimes \mathcal{H}) \simeq \mathcal{L}(\mathcal{L}(\mathcal{H}, \mathcal{H}), \mathcal{L}(\mathcal{H}, \mathcal{H}))$ explicit. The spectrum of $K$ thus agrees with the eigenvalues of the linear map $T$ acting on matrices by

$$T(M) = \text{Tr}_2\left[O_\mu(\text{id} \otimes M)\right]. \tag{28}$$

We claim that there is an eigendecomposition

$$T(M) = \sum \gamma_i A_i \text{Tr}\left[A_i M\right] \tag{29}$$

where $A_i$ are orthonormal hermitian matrices. Moreover, the eigenfunctions of $K$ are $f_i(x) = \text{Tr}\left[\rho(x)A_i\right]$. This result follows from standard results in linear algebra, we give all details in the next subsection.

## C.4 Spectral decomposition of linear maps preserving hermitian matrices

We consider the space of matrices $\mathbb{C}^{n \times n}$ equipped with the usual scalar product $\langle A, B \rangle = \text{Tr}\left[A^\dagger B\right]$ which agrees with the standard scalar product on $C^{n^2}$ after vectorisation. We will need them following fact: For hermitian matrices $A, B$ the scalar product $\langle A, B \rangle \in \mathbb{R}$ is real.

**Lemma 3.** *Let $T : \mathbb{C}^{n \times n} \to \mathbb{C}^{n \times n}$ be a linear and hermitian map that maps hermitian matrices to hermitian matrices. Then there is a eigendecomposition $(\gamma_i, H_i)$ with real eigenvalues $\gamma_i$ and orthonormal hermitian matrices $H_i$ such that*

$$T(A) = \sum_i \gamma_i H_i \text{Tr}\left[H_i^\dagger A\right]. \tag{30}$$

*Proof.* Hermitian matrices can be diagonalized with real values $\gamma_i$ so we can write

$$T(A) = \sum_i \gamma_i X_i \text{Tr}\left[X_i^\dagger A\right] \tag{31}$$

where $X_i$ form an orthonormal eigenbasis. It remains to show that we can find such a decomposition where the $X_i$ are hermitian. We decompose $X_i = \tilde{H}_i + i\tilde{S}_i$ where $\tilde{H}_i$ and $\tilde{S}_i$ are hermitian. Then we observe

$$\gamma_i(\tilde{H}_i + i\tilde{S}_i) = \gamma_i X_i = T(X_i) = T(\tilde{H}_i) + iT(\tilde{S}_i). \tag{32}$$

Using the invariances of $T$ on hermitian matrices we conclude that $\tilde{S}_i$ and $\tilde{H}_i$ are again eigenvectors with eigenvalue $\gamma_i$. Now we can iteratively replace $X_i$ by either $\tilde{S}_i$ or $\tilde{H}_i$ so that the set of vectors remains a basis. Finally we orthonormalize the resulting basis of all eigenspaces using the Gram-Schmidt procedure. Since scalar products of hermitian matrices are real we obtain an orthonormal eigenbasis $H_i$ consisting of hermitian matrices. $\square$

We now apply this to the integral operator for the quantum embedding. Recall that the linear map $T$ acting on matrices was defined by

$$T(M) = \text{Tr}_2\left[O_\mu(\text{id} \otimes M)\right]. \tag{33}$$

Clearly, $T$ is linear. To show that $T$ is hermitian we observe that

$$\langle M, T(M) \rangle = \text{Tr}\left[M^\dagger \text{Tr}_2\left[O_\mu(\text{id} \otimes M)\right]\right] = \int \text{Tr}\left[M^\dagger \text{Tr}_2\left[\rho(y) \otimes \rho(y)(\text{id} \otimes M)\right]\right]\mu(\text{d}y)$$

$$= \int \text{Tr}\left[M^\dagger \rho(y)\right]\text{Tr}\left[\rho(y)M\right]\mu(\text{d}y) \in \mathbb{R}. \tag{34}$$

Similarly we see that $T$ preserves hermitian matrices, indeed, if $M = M^\dagger$

$$T(M) = \int \mathrm{Tr}_2\left[\rho(y) \otimes \rho(y)(\mathrm{id} \otimes M)\right]\mu(\mathrm{d}y) = \int \rho(y)\mathrm{Tr}\left[\rho(y)M\right]\mu(\mathrm{d}y). \tag{35}$$

which is hermitian because $\rho(y)$ is hermitian and the scalar product of hermitian matrices is real. Using Lemma 3 above we conclude that we can write $T(M) = \sum_i \gamma_i A_i \mathrm{Tr}\left[A_i M\right]$ where $\gamma_i$ are the eigenvalues of $T$ which agree with the eigenvalues of the corresponding integral operator and the eigenfunctions are given by $x \to \mathrm{Tr}\left[\rho(x)A_i\right]$.

## C.5 A complete example

To illustrate the analysis above we consider the setting from Example 2 where $x \to \cos(x/2)|0\rangle + i\sin(x/2)|1\rangle$. Then $\tilde{\mathcal{F}} = \langle \sin(x), \cos(x)\rangle$ and $\mathcal{F} = \langle \sin^2(x), \cos^2(x), \sin(x)\cos(x)\rangle$. Note that the RKHS has dimension 4 when the relative phase between $a(x)$ and $b(x)$ is not constant (then $\bar{a}b$ and $a\bar{b}$ are not linearly dependent). The feature map of the physical kernel for our example is

$$\rho(y) = \begin{pmatrix} \cos^2(\frac{y}{2}) & -i\cos(\frac{y}{2})\sin(\frac{y}{2}) \\ i\cos(\frac{y}{2})\sin(\frac{y}{2}) & \sin^2(\frac{y}{2}) \end{pmatrix}. \tag{36}$$

For the analysis of the integral operator we need the matrix elements of the linear map $T$ We observe that in index notation using the Einstein summation convention and denoting complex conjugation without transposition by $*$

$$T(M)_{ij} = \int \rho(y)_{ij}\rho(y)_{kl}M_{lk}\,\mu(\mathrm{d}y) = \int \rho(y)_{ij}\rho^*(y)_{lk}M_{lk}\,\mu(\mathrm{d}y). \tag{37}$$

Using vectorisation we obtain

$$\mathrm{Vec}(T(M)) = \int \mathrm{Vec}(\rho(y))\mathrm{Vec}(\rho(y))^\top \mu(\mathrm{d}y)\mathrm{Vec}(M) = A_\mu \mathrm{Vec}M. \tag{38}$$

In our example we obtain

$$
\begin{aligned}
A_\mu &= \frac{1}{\pi}\int_0^\pi \begin{pmatrix} \cos^2(\frac{y}{2}) \\ -i\cos(\frac{y}{2})\sin(\frac{y}{2}) \\ i\cos(\frac{y}{2})\sin(\frac{y}{2}) \\ \sin^2(\frac{y}{2}) \end{pmatrix} \left(\cos^2(\frac{y}{2}) \quad i\cos(\frac{y}{2})\sin(\frac{y}{2}) \quad -i\cos(\frac{y}{2})\sin(\frac{y}{2}) \quad \sin^2(\frac{y}{2})\right)\mathrm{d}y \\
&= \frac{1}{\pi}\int_0^\pi \begin{pmatrix} \cos^4(\frac{y}{2}) & i\cos^3(\frac{y}{2})\sin(\frac{y}{2}) & -i\cos^3(\frac{y}{2})\sin(\frac{y}{2}) & \cos^2(\frac{y}{2})\sin^2(\frac{y}{2}) \\ -i\cos^3(\frac{y}{2})\sin(\frac{y}{2}) & \cos^2(\frac{y}{2})\sin^2(\frac{y}{2}) & -\cos^2(\frac{y}{2})\sin^2(\frac{y}{2}) & -i\cos(\frac{y}{2})\sin^3(\frac{y}{2}) \\ i\cos^3(\frac{y}{2})\sin(\frac{y}{2}) & -\cos^2(\frac{y}{2})\sin^2(\frac{y}{2}) & \cos^2(\frac{y}{2})\sin^2(\frac{y}{2}) & i\cos(\frac{y}{2})\sin^3(\frac{y}{2}) \\ \cos^2(\frac{y}{2})\sin^2(\frac{y}{2}) & i\cos(\frac{y}{2})\sin^3(\frac{y}{2}) & -i\cos(\frac{y}{2})\sin^3(\frac{y}{2}) & \sin^4(\frac{y}{2}) \end{pmatrix}\mathrm{d}y \\
&= \frac{1}{8}\begin{pmatrix} 3 & 0 & 0 & 1 \\ 0 & 1 & -1 & 0 \\ 0 & -1 & 1 & 0 \\ 1 & 0 & 0 & 3 \end{pmatrix}
\end{aligned}
\tag{39}
$$

We obtain the eigenvalues $\frac{1}{2}, \frac{1}{4}, \frac{1}{4}, 0$ the eigenvectors are, in matrix notation,

$$H_1 = \begin{pmatrix} 1 & 0 \\ 0 & 1 \end{pmatrix}, \quad H_2 = \begin{pmatrix} 1 & 0 \\ 0 & -1 \end{pmatrix}, \quad H_3 = \begin{pmatrix} 0 & i \\ -i & 0 \end{pmatrix}, \quad H_4 = \begin{pmatrix} 0 & 1 \\ 1 & 0 \end{pmatrix}. \tag{40}$$

The corresponding eigenfunctions $f_i$ of the integral operator are given by $x \to \mathrm{Tr}\left[\rho(x)H_i\right]$, i.e.,

$$
\begin{aligned}
f_1(x) &= 1, & f_2(x) &= \cos^2(\tfrac{x}{2}) - \sin^2(\tfrac{x}{2}) = \cos(x), \\
f_3(x) &= 2\cos(\tfrac{x}{2})\sin(\tfrac{x}{2}) = \sin(x), & f_4(x) &= 0.
\end{aligned}
\tag{41}
$$

We can also parametrize the functions in the RKHS by $a\cos(x+b) + c$ with $a, b, c \in \mathbb{R}$.

Let us also look at the generalization to the vector valued case with $d$-qubits. Then the RKHS is given by all functions of the form

$$x \to \prod_{i=1}^d (a_i\cos(x_i + b_i) + c_i). \tag{42}$$

The eigenfunctions of the integral operator are given by

$$\prod_{i=1}^{d} \sin^{\alpha_i}(x_i) \cos^{\beta_i}(x_i) \tag{43}$$

where $\alpha_i, \beta_i$ are non-negative integers satisfying $\alpha_i + \beta_i \leq 1$. The corresponding eigenvalue is $2^{-d-\sum(\alpha_i+\beta_i)}$. The degeneracy of the eigenvalue $2^{-d-l}$ can be calculated to $2^l \binom{d}{l}$.

## D    Proof of Theorem 1

In this section we prove Theorem 1 which will follow easily from the result below. We remark that the following theorem is by no means sharp but a detailed analysis when learning is not possible is of limited interest. Note that again typical lower bounds for the learning performance are focused on the case $n \to \infty$ [43].

**Theorem 3.** *Consider a measure space* $(X, \mu)$ *such that* $\mu(X) = 1$ *with a kernel* $k$ *satisfying* $k(x,x) = 1$ *for all* $x \in X$. *Denote by* $\gamma_{max}$ *the largest eigenvalue of the corresponding integral operator. Suppose we have* $n$ *training points* $\mathcal{D}_n = \{(x_i, y_i), 1 \leq i \leq n\}$ *with* $(x_i, y_i) \in X \times \mathbb{R}$ *where* $x_i$ *are i.i.d. draws from* $\mu$ *and* $y_i = f(x_i)$ *for some square integrable function* $f$. *Then, for any* $\varepsilon > 0$ *with probability at least* $1 - \varepsilon - \gamma_{max} n^4$

$$\|f - \hat{f}_n^\lambda\|_2 \geq \left(1 - \sqrt{\frac{2\gamma_{max}n^2}{\varepsilon}}\right) \|f\|_2 \tag{44}$$

*for all* $\lambda \geq 0$ *where* $\hat{f}_n^\lambda$ *denotes the kernel ridge regression estimator for training data* $(x_i, y_i)$.

*Proof.* Denote the eigenvalues of the integral operator by $\gamma_i$ with $\gamma_1 = \gamma_{max}$. Standard results for integral operators imply

$$\sum_i \gamma_i = \int k(x,x)\,\mu(\mathrm{d}x) = 1 \tag{45}$$

$$\sum_i \gamma_i^2 = \int k(x,y)^2\,\mu(\mathrm{d}x)\mu(\mathrm{d}y) = \|k\|_2^2. \tag{46}$$

We conclude that

$$\|k\|_2^2 = \sum_i \gamma_i^2 \leq \gamma_{max} \sum_i \gamma_i = \gamma_{max}. \tag{47}$$

Since $\mathbb{E}_{\mu\otimes\mu}\left[k(x,y)^2\right] = \|k\|_2^2$, Markov's inequality together with (47) implies $\mathbb{P}_{\mu\otimes\mu}(|k(x,y)| \geq \varepsilon) \leq \frac{\gamma_{max}}{\varepsilon^2}$. Let $A_n = \{|k(x_i, x_j)| \leq \frac{1}{2n}$ for all $i \neq j\}$. Using the union bound we conclude that

$$\mathbb{P}_{\mathcal{D}_n}(A_n) \geq 1 - n^2 \mathbb{P}_{\mu\otimes\mu}\left(|k(x,y)| \geq \frac{1}{2n}\right) \geq 1 - 4n^4 \gamma_{max}. \tag{48}$$

Conditioned on $A_n$ we can bound the eigenvalues of the kernel matrix $K(X,X)_{i,j} = k(x_i, x_j)$ using Gerschgorin circles by $1 - n\frac{1}{2n} = \frac{1}{2}$ and thus

$$K(X,X)^{-1} \leq 2 \cdot \mathrm{id}_n. \tag{49}$$

Let us denote the Mercer decomposition of $k$ by

$$k(x,y) = \sum_i \gamma_i f_i(x) f_i(y) \tag{50}$$

where $f_i$ are the orthonormal eigenfunctions. Then we can bound

$$|k(x,\cdot)|_2^2 = \int k(x,y)^2\,\mu(\mathrm{d}y) = \int \sum_i \gamma_i f_i(x) f_i(y) \sum_j \gamma_j f_j(x) f_j(y)\,\mu(\mathrm{d}y)$$
$$= \sum_{i,j} \delta_{ij} \gamma_i \gamma_j f_i(x) f_j(x) \leq \gamma_{max} \sum_i \gamma_i f_i(x)^2 = \gamma_{max}. \tag{51}$$

The kernel ridge regression function $f_n^\lambda$ can be written as

$$f_n^\lambda = \sum_i \alpha_i k(x_i, \cdot) \tag{52}$$

where the vector $\alpha$ is given by $\alpha = (K(X,X) + \lambda \operatorname{id}_{n\times n})^{-1} y$ with $y \in \mathbb{R}^n$ denoting the vector with components $y_i$. Using (49) we conclude that conditioned on $A_n$ we have

$$\|\alpha\|^2 \le 2\|y\|^2. \tag{53}$$

We now claim that for any $\varepsilon > 0$ with probability $1 - \varepsilon$ we have

$$|y|^2 \le \tfrac{n}{\varepsilon} \|f\|_2^2. \tag{54}$$

To show this we remark that $\mathbb{E}(y_i^2) = \mathbb{E}(f(x_i)^2) = \|f\|_2^2$ because we assumed that $x_i$ is i.i.d. with distribution $\mu$. Using Markov's inequality (and $|y|^2 \ge 0$) we can bound

$$\mathbb{P}\left(|y|^2 \ge \frac{n}{\varepsilon}\|f\|_2^2\right) \le \mathbb{E}\left(\frac{|y|^2}{n\varepsilon^{-1}\|f\|_2^2}\mathbf{1}_{|y|^2 \ge n\varepsilon^{-1}\|f\|_2^2}\right) \le \frac{\varepsilon}{n\|f\|_2^2}\mathbb{E}\left(\sum_{i=1}^n |y_i|^2\right) = \varepsilon. \tag{55}$$

This implies the claim (54).

Using (51), (53), and (54) we conclude that the $L^2$ norm of $f_n^\lambda$ satisfies now with probability $1 - \varepsilon - \gamma_{max}n^4$ the bound

$$\|f_n^\lambda\|_2 \le \sum_i |\alpha_i|\|k(x_i,\cdot)\|_2 \le \sqrt{\gamma_{max}}\sum_i |\alpha_i| \le \sqrt{\gamma_{max}n}\sqrt{\sum_i \alpha_i^2}$$

$$\le \sqrt{\gamma_{max}n}\sqrt{\frac{2n\|f\|_2^2}{\varepsilon}} \le \sqrt{\frac{2\gamma_{max}n^2}{\varepsilon}}\|f\|. \tag{56}$$

We conclude that with probability $1 - \varepsilon - \gamma_{max}n^4$

$$\|f - f_n^\lambda\|_2 \ge \|f\|_2 - \|f_n^\lambda\|_2 \ge \|f\|_2\left(1 - \sqrt{\frac{2\gamma_{max}n^2}{\varepsilon}}\right). \tag{57}$$

$\square$

The proof of Theorem 1 is now a consequence of the result above.

*Proof of Theorem 1.* The general strategy of the proof is to show that the result follows from Theorem 3 for sufficiently large $d$. We first note that, using the assumption $\mu = \bigotimes \mu_i$

$$\rho_\mu = \bigotimes \rho_{\mu_i} \tag{58}$$

and thus

$$\operatorname{Tr}[\rho_\mu] = \prod \operatorname{Tr}[\rho_{\mu_i}] \le \delta^d. \tag{59}$$

Lemma 1 then implies that the largest eigenvalue of the integral operator is bounded by $\gamma_{max}(d) \le \delta^{d/2}$. Next we observe that there is $d_0(\delta, l, \varepsilon)$ such that for $d \ge d_0$

$$\delta^{d/2} \le \varepsilon d^{-4l}/2 \qquad \text{and} \qquad \delta^{d/2} \le \varepsilon^3 d^{-2l}/4, \tag{60}$$

because the left sides of the equations are decaying exponentially in $d$ (recall that $\delta < 1$) and the right sides only polynomially.

Using the estimates above and the assumption $n \le d^l$ we conclude that for $d \ge d_0$

$$\gamma_{max} \le \delta^{d/2} \le \varepsilon d^{-4l}/2 \le \varepsilon n^{-4}/2 \quad \Rightarrow \gamma_{max}n^4 \le \varepsilon/2 \tag{61}$$

$$\gamma_{max} \le \delta^{d/2} \le \varepsilon^3 d^{-2l}/2 \le \varepsilon^3 n^{-2}/4 \quad \Rightarrow \sqrt{4\gamma_{max}n^2\varepsilon^{-1}} \le \varepsilon. \tag{62}$$

We now denote the $\varepsilon$ used in Theorem 3 as $\varepsilon'$ and set $\varepsilon' = \varepsilon/2$. Theorem 3 and (61) and (62) then imply that with probability at least

$$1 - \varepsilon' - \gamma_{max}n^4 \ge 1 - \varepsilon/2 - \varepsilon/2 = 1 - \varepsilon \tag{63}$$

the bound

$$\|f - \hat{f}_n^\lambda\|_2^2 \ge \left(1 - \sqrt{\frac{2\gamma_{max}n^2}{\varepsilon'}}\right)\|f\|_2 = \left(1 - \sqrt{\frac{4\gamma_{max}n^2}{\varepsilon}}\right)\|f\|_2 \ge (1-\varepsilon)\|f\|_2 \tag{64}$$

holds for all $\lambda \ge 0$. This completes the proof. $\square$

# E  Proof of Theorem 2

We introduce some theory and notation necessary for the proof. We investigate the behavior of reduced density matrices when $V$ is distributed according to the Haar-measure on the group of unitary matrices. The first even moments of the Haar measure on $U(2^d)$ are given by (see e.g., [53])

$$\int V_{ij} V^*_{i'j'}\, \mu(\mathrm{d}V) = \frac{\delta_{ii'}\delta_{jj'}}{2^d}$$

$$\int V_{i_1 j_1} V_{i_2 j_2} V^*_{i'_1 j'_1} V^*_{i'_2 j'_2}\, \mu(\mathrm{d}V) = \frac{1}{2^{2d}-1}\left(\delta_{i_1 i'_1}\delta_{j_1 j'_1}\delta_{i_2 i'_2}\delta_{j_2 j'_2} + \delta_{i_1 i'_2}\delta_{j_1 j'_2}\delta_{i_2 i'_1}\delta_{j_2 j'_1}\right)$$
$$- \frac{1}{2^d(2^{2d}-1)}\left(\delta_{i_1 i'_1}\delta_{j_1 j'_2}\delta_{i_2 i'_2}\delta_{j_2 j'_1} + \delta_{i_1 i'_2}\delta_{j_1 j'_1}\delta_{i_2 i'_1}\delta_{j_2 j'_2}\right).$$
(65)

Note that here and in the following $V^*$ the conjugated (but not transposed) matrix. Let us remark that while random circuits that output Haar-distributed unitaries require an exponential (in $d$) number of gates our arguments actually only require unitary $t$-designs which are point distributions that match the first $t$ moments of the Haar measure. In particular a 2-design is a measure with finite support on unitary matrices satisfying (65) (and odd moments of lower order vanish). Those can be implemented using polynomially many gates. For details and further information we refer to the literature [54].

Recall the definition of the projected quantum kernel

$$\tilde{\rho}_m^V(x) = \mathrm{Tr}_{m+1\ldots d}\left[\rho^V(x)\right].$$
(66)

To denote the partial trace in index notation we split the index $i \in \{1,\ldots,2^d\}$ in $(\alpha, \bar{\alpha})$ where $\alpha \in \{1,\ldots,2^m\}$ denotes the index corresponding to the first $m$ qubits and $\bar{\alpha} \in \{1,\ldots,2^{d-m}\}$ denotes the index corresponding to the remaining $d-m$ qubits. We will always use roman letters for indices in $\{1,\ldots,2^d\}$, greek letters for indices in $\{1,\ldots,2^m\}$ and greek letters with a bar for indices in $\{1,\ldots,2^{d-m}\}$. In particular, summing 1 over $\bar{\alpha}$ results in $2^{d-m}$ and summing over $i$ results in $2^d$. We will always use Einstein summation convention in the following so that, e.g. $\delta_{\bar{\alpha}\bar{\alpha}} = 2^{d-m}$. We are now ready to prove Theorem 2.

*Proof of Theorem 2.* We start to prove the asymptotic expression for the reduced density matrix which is a standard result. We can write

$$\mathbb{E}_V\left[\tilde{\rho}_m^V(x)_{\alpha_1,\alpha_2}\right] = \mathbb{E}_V\left[V_{\alpha_1\bar{\alpha},j}\rho(x)_{j,j'}V^*_{\alpha_2\bar{\alpha},j'}\right] = \frac{2^{d-m}}{2^d}\delta_{\alpha_1\alpha_2}\delta_{jj'}\rho(x)_{j,j'} = 2^{-m}\delta_{\alpha_1\alpha_2}\mathrm{Tr}\left[\rho(x)\right].$$
(67)

To show the concentration around the expectation value we need to calculate the variance of this expression. We calculate the second moment of the reduced density matrix

$$\mathbb{E}_V\left[\tilde{\rho}_m^V(x)_{\alpha_1,\alpha_2}\tilde{\rho}_m^V(y)_{\beta_1,\beta_2}\right] = \mathbb{E}_V\left[V_{\alpha_1\bar{\alpha},j_1}\rho(x)_{j_1,j'_1}V^*_{\alpha_2\bar{\alpha},j'_1}V_{\beta_2\bar{\beta},j_2}\rho(y)_{j_2,j'_2}V^*_{\beta_1\bar{\beta},j'_2}\right]$$
$$= A_1 + A_2 + A_3 + A_4.$$
(68)

Here the terms $A_i$ correspond to the four contributions on the right hand side of (65). The four terms can be evaluated to (assuming that $\text{Tr}\left[\rho(x)\right] = 1$ for all $x$)

$$A_1 = \frac{1}{2^{2d}-1}\delta_{\alpha_1\alpha_2}2^{d-m}\text{Tr}\left[\rho(x)\right]\delta_{\beta_1\beta_2}2^{d-m}\text{Tr}\left[\rho(y)\right] = \frac{2^{2d}}{2^{2d}-1}2^{-2m}\delta_{\alpha_1\alpha_2}\delta_{\beta_1\beta_2}$$

$$= 2^{-2m}\delta_{\alpha_1\alpha_2}\delta_{\beta_1\beta_2} + \frac{1}{2^{2d}-1}2^{-2m}\delta_{\alpha_1\alpha_2}\delta_{\beta_1\beta_2}$$

$$A_2 = \frac{1}{2^{2d}-1}\delta_{\alpha_1\beta_2}\delta_{\bar{\alpha}\bar{\beta}}\delta_{j_1 j'_2}\delta_{\alpha_2\beta_1}\delta_{\bar{\beta}\bar{\alpha}}\delta_{j_2 j'_1}\rho(x)_{j_1,j'_1}\rho(y)_{j_2,j'_2}$$

$$= \frac{1}{2^{2d}-1}\delta_{\bar{\alpha}\bar{\alpha}}\rho(x)_{j_1,j'_1}\rho(y)_{j'_1,j_1}\delta_{\alpha_1\beta_2}\delta_{\alpha_2\beta_1} = \frac{2^{d-m}}{2^{2d}-1}\text{Tr}\left[\rho(x)\rho(y)\right]\delta_{\alpha_1\beta_2}\delta_{\alpha_2\beta_1}$$

(69)

$$A_3 = -\frac{1}{2^d(2^{2d}-1)}\delta_{\alpha_1\alpha_2}\delta_{\bar{\alpha}\bar{\alpha}}\delta_{j_1 j'_2}\delta_{\beta_1\beta_2}\delta_{\bar{\beta}\bar{\beta}}\delta_{j_2 j'_1}\rho(x)_{j_1,j'_1}\rho(y)_{j_2,j'_2}$$

$$= -\frac{2^{2d-2m}}{2^d(2^{2d}-1)}\rho(x)_{j_1,j'_1}\rho(y)_{j'_1,j_1}\delta_{\alpha_1\alpha_2}\delta_{\beta_1\beta_2} = -\frac{2^{d-2m}}{2^{2d}-1}\text{Tr}\left[\rho(x)\rho(y)\right]\delta_{\alpha_1\alpha_2}\delta_{\beta_1\beta_2}$$

$$A_4 = -\frac{1}{2^d(2^{2d}-1)}\delta_{\alpha_1\beta_2}\delta_{\bar{\alpha}\bar{\beta}}\delta_{j_1 j'_1}\delta_{\alpha_2\beta_1}\delta_{\bar{\alpha}\bar{\beta}}\delta_{j_2 j'_2}\rho(x)_{j_1,j'_1}\rho(y)_{j_2,j'_2}$$

$$= -\frac{1}{2^d(2^{2d}-1)}\delta_{\bar{\alpha}\bar{\alpha}}\rho(x)_{j_1,j_1}\rho(y)_{j_2,j_2}\delta_{\alpha_1\beta_2}\delta_{\alpha_2\beta_1} = -\frac{2^{-m}}{2^{2d}-1}\delta_{\alpha_1\beta_2}\delta_{\alpha_2\beta_1}$$

Altogether we obtain

$$\mathbb{E}_V\left[\tilde{\rho}_m^V(x)_{\alpha_1,\alpha_2}\tilde{\rho}_m^V(y)_{\beta_1,\beta_2}\right] = \delta_{\alpha_1\alpha_2}\delta_{\beta_1\beta_2}2^{-2m}\left(1 - 2^{-d}\text{Tr}\left[\rho(x)\rho(y)\right] + 2^{-2d}\right)$$
$$+ \delta_{\alpha_1\beta_2}\delta_{\alpha_2\beta_1}2^{-m}\left(2^{-d}\text{Tr}\left[\rho(x)\rho(y)\right] - 2^{-2d}\right) + O(2^{-3d})$$

(70)

Recall that the complex variance of a random variable is defined by $\mathbb{E}\left[|X|^2\right] - |\mathbb{E}\left[X\right]|^2$. Using (70) and that $\tilde{\rho}$ is hermitian we can bound the variance of the entries of $\tilde{\rho}(x)$ by

$$\mathbb{E}_V\left[\tilde{\rho}_m^V(x)_{\alpha,\beta}(\tilde{\rho}_m^V(x)_{\alpha,\beta})^*\right] - \mathbb{E}_V\left[\tilde{\rho}_m^V(x)_{\alpha,\beta}\right]\mathbb{E}_V\left[(\tilde{\rho}_m^V(x)_{\alpha,\beta})^*\right]$$
$$= \mathbb{E}_V\left[\tilde{\rho}_m^V(x)_{\alpha,\beta}\tilde{\rho}_m^V(x)_{\beta,\alpha}\right] - \mathbb{E}_V\left[\tilde{\rho}_m^V(x)_{\alpha,\beta}\right]\mathbb{E}_V\left[\tilde{\rho}_m^V(x)_{\beta,\alpha}\right]$$

(71)

$$= 2^{-2m}\delta_{\alpha\beta} - (2^{-m})^2\delta_{\alpha\beta} + O(2^{-d}) = O(2^{-d}).$$

This shows that $\tilde{\rho}^V(x)$ is close to $2^{-m}\text{id}$ with high probability for large $d$ and finishes the proof of the first part of the theorem.

We now turn to the evaluation of the averaged operator $\mathbb{E}_V\left[O_\mu\right]$ and the corresponding operator

$$T(M) = \text{Tr}_2\left[\mathbb{E}_V\left[O_\mu\right](\text{id} \otimes M)\right]$$ (72)

whose matrix elements we denote by $T_{\alpha_1\alpha_2;\beta_1\beta_2}$ so that $T(M)_{\alpha_1,\alpha_2} = T_{\alpha_1\alpha_2;\beta_1\beta_2}M_{\beta_1,\beta_2}$. We assume that $\rho(x)$ is pure for all $x$, i.e., $\text{Tr}\left[\rho(x)^2\right] = 1$. We have seen in (37) that the matrix elements of this operator are given by

$$\mathbb{E}_V\left[\int \tilde{\rho}_m^V(y) \otimes (\tilde{\rho}_m^V(y))^* \, \mu(\mathrm{d}y)\right] = \int \mathbb{E}_V\left[\tilde{\rho}_m^V(y) \otimes (\tilde{\rho}_m^V(y))^*\right]\mu(\mathrm{d}y).$$ (73)

From (70) we obtain for the matrix elements

$$\mathbb{E}_V\left[\tilde{\rho}^V(y)_{\alpha_1,\alpha_2}(\tilde{\rho}^V(y)_{\beta_1,\beta_2})^*\right] = \mathbb{E}_V\left[\tilde{\rho}^V(y)_{\alpha_1,\alpha_2}\tilde{\rho}^V(y)_{\beta_2,\beta_1}\right]$$

$$= 2^{-2m}\delta_{\alpha_1\alpha_2}\delta_{\beta_1\beta_2} + \frac{2^{-m}}{2^d}\delta_{\alpha_1\beta_1}\delta_{\alpha_2\beta_2} - \frac{2^{-2m}}{2^d}\delta_{\alpha_1\alpha_2}\delta_{\beta_1\beta_2} + O(2^{-2d}).$$

(74)

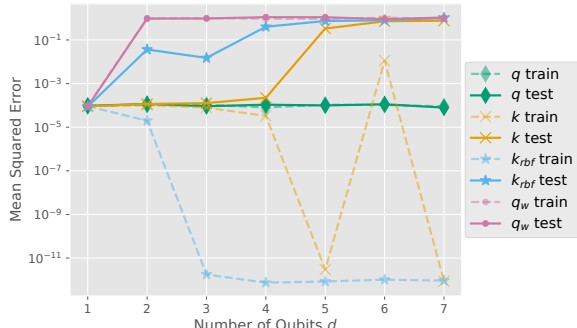

Figure 4: Similar as in Fig. 2. However, for the full quantum kernel $k$ and the rbf kernel, we compute train and test loss over multiple choices of the regularization parameter. For each number of qubits, we only report the loss of the method that achieved smallest test loss. Note that, although this is invalid to asses the power of the full and rbf kernel, it shows, that the poor performance is not due to the choice of regularization. Since we cherry-pick on the test loss, it can happen that an underfitting regularization has the best test loss, which explains the outlier on $k$ at $d = 6$.

Since this is independent of $y$ we can write the matrix elements of $T$ as

$$T_{\alpha_1\alpha_2;\beta_1\beta_2} = 2^{-2m}(1 - 2^{-d})\delta_{\alpha_1\alpha_2}\delta_{\beta_1\beta_2} + \frac{2^{-m}}{2^d}\delta_{\alpha_1\beta_1}\delta_{\beta_2\alpha_2} + O(2^{-2d}) \tag{75}$$

From here we conclude that $T$ can be written as

$$T(M)_{\alpha_1,\alpha_2} = 2^{-2m}(1 - 2^{-d})\delta_{\alpha_1\alpha_2}\delta_{\beta_1\beta_2}M_{\beta_1,\beta_2} + \frac{2^{-m}}{2^d}\delta_{\alpha_1\beta_1}\delta_{\alpha_2\beta_2}M_{\beta_1,\beta_2} + O(2^{-2d})$$

$$= 2^{-2m}(1 - 2^{-d})\delta_{\alpha_1\alpha_2}M_{\beta_1,\beta_1} + \frac{2^{-m}}{2^d}M_{\alpha_1,\alpha_2} + O(2^{-2d}) \tag{76}$$

or, more concisely,

$$T(M) = \frac{2^{-m}}{2^d}M + 2^{-2m}(1 - 2^{-d})\mathrm{id}_{2^m \times 2^m}\mathrm{Tr}\left[\mathrm{id}_{2^m \times 2^m}M\right] + O(2^{-2d}) \tag{77}$$

and we observe that $T$ is the sum of a multiple of the identity and a rank one perturbation (plus higher order terms): In particular the eigenvalues neglecting the perturbation are

$$\gamma_1 = 2^{-2m}(1 - 2^{-d})\mathrm{Tr}\left[\mathrm{id}_{2^m \times 2^m}\mathrm{id}_{2^m \times 2^m}\right] + 2^{-m-d} = 2^{-m}(1 - 2^{-d}) + 2^{-m-d} = 2^{-m} \tag{78}$$

with eigenvector $M_1 = \mathrm{id}_{2^m \times 2^m}$ and $\gamma_2 = \ldots = \gamma_{2^m \times 2^m} = 2^{-m-d}$ with traceless eigenvectors, i.e., $\mathrm{Tr}\left[\mathrm{id}_{2^m \times 2^m}M_i\right] = 0$ for $i \neq 1$. Standard bounds show that the higher order terms change the eigenvalues only by a term of order $O(2^{-2d})$. Finally, we observe that the function mapping $x \to \mathrm{Tr}\left[\tilde{\rho}_m^V(x)M_1\right]$ is a constant function for any $V$. Indeed,

$$\mathrm{Tr}\left[\tilde{\rho}_m^V(x)M_1\right] = \mathrm{Tr}\left[\mathrm{Tr}_{m+1\ldots d}\left[V\rho(x)V^\dagger\right]\right] = \mathrm{Tr}\left[V\rho(x)V^\dagger\right] = \mathrm{Tr}\left[\rho(x)\right] = 1. \tag{79}$$

$\square$

## F  More on experiments

For details on the implementation we refer to the provided code.[5] We emphasize that our experiments simulate the full quantum state and thus work with the true values of the quantum kernel. This is an idealized setting and neglects the effect of finite measurements. Please see our discussion on Barren Plateaus in the main paper.

To reduce computations and speed-up the simulation, we compute the full quantum kernel $k(x, x') = \prod \cos^2\left((x_i - x_i')/2\right)$ directly without simulating a quantum circuit. For the biased kernels we

---
[5]https://github.com/jmkuebler/quantumbias

recommend (and implement it that way) to completely simulate $\rho_1^V(x_i)$ for all $i = 1, \ldots, n$ and store the reduced density matrices ($2 \times 2$ hermitian matrices). On a real quantum device this would correspond to doing quantum state tomography [36]. The benefit of this is that we only need to simulate the quantum circuit $n$ times and can then directly compute the biased kernels via matrix products and tracing. If we chose to compute each entry of the kernel matrix individually we would have to simulate the circuit $n^2$ times.

**Random generation of $V$.**   In order to generate random unitary matrices $V$ we use the PennyLane function RANDOMLAYERS [51]. For $d$ qubits we use $d^2$ layers of single qubit rotations and 2-qubit entangling gates. For more details and the used seeds please refer to the provided implementation.

**Choice of regularization.**   For the biased kernels $q, q_w$ regularization does not matter much, since they have only a four-dimensional RKHS and we consider sample sizes much larger than that. The RKHS simply does not have enough capacity to overfit to random noise. We therefore set the regularization $\lambda = 0$ for the biased kernels. On the other hand for the higher dimensional kernels $k, k_{\text{rbf}}$, the regulariyation strongly influences their performance. For the experiment in the main paper we set $\lambda = 10^{-3}$ for the latter methods. Note that in a real application one should use cross-validation or other model selection techniques to find good hyperparameters, which we omitted for simplicity. To exclude that the bad performance of $k$ and $k_{\text{rbf}}$ stems from a bad choice of regularization, we include experiments where we fit kernel ridge regression for 15 values of $\lambda$ on a logarithmic grid from $10^{-6}$ to $10^4$. We then cherry-pick only the solution that performs best and report it in Figure 4. Note that such an approach is of course not legit to asses the actual performance. However, it serves to bound the performance for the optimal choice of regularization. Our observations show that the behavior does not significantly change and we conclude that the performance difference indeed comes from the spectral bias as predicted by our theory.

**Additional experiments.**   To show how the kernel target alignment changes as we increase the number of qubits $d$, we include further histograms in Figure 5. The estimated kernel alignment correlates with the learning performance reported in Figure 2.

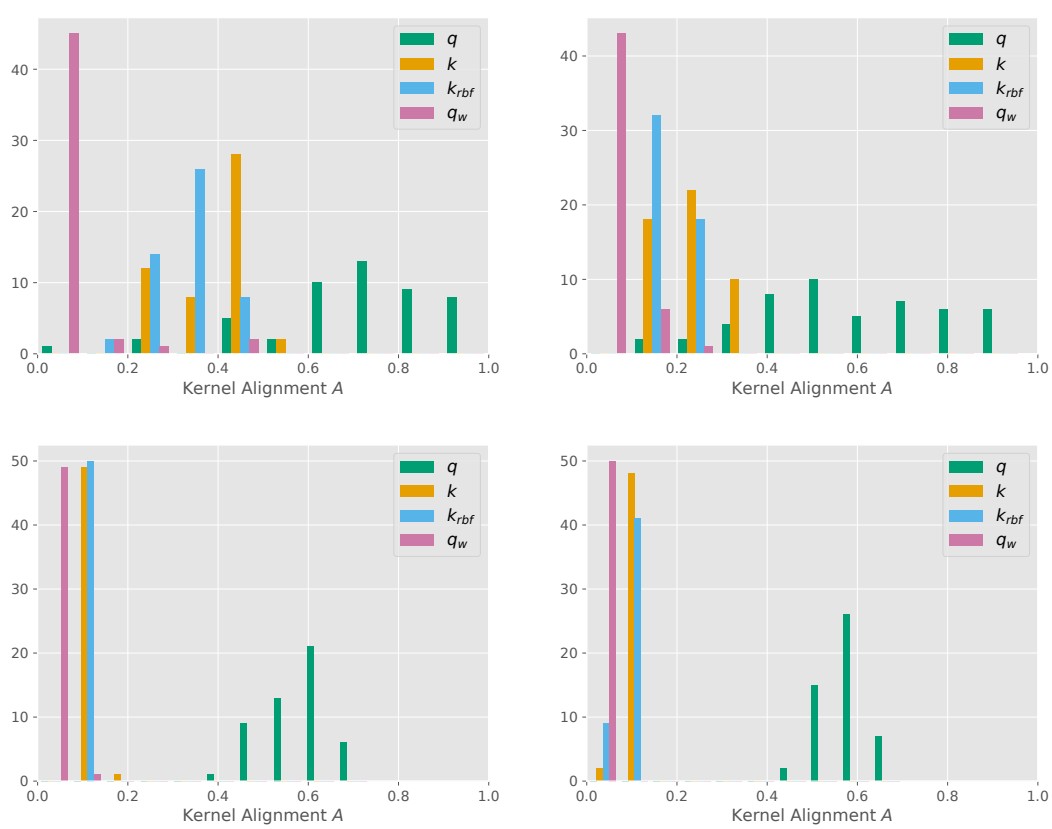

Figure 5: Kernel Target Alignment for $d = 1, 3, 5, 7$.