# OpenReview forum: "The Inductive Bias of Quantum Kernels"
_NeurIPS.cc/2021/Conference — NeurIPS 2021 Poster_

### Official Review · Reviewer_fXtY · 2021-07-13

**Rating:** 8
**Confidence:** 3

**Summary:**

The paper investigates quantum kernels and learning with them, and whether it can be advantageous over using classical computations. A lemma is proposed which bounds the eigenvalues of the integral operator of a quantum kernel, and a theorem which shows how the empirical risk of the KRR solution with certain kinds of quantum kernels is bounded from below. Further, a biased quantum kernel is introduced for which the integral operators are also investigated. From these results (and experimentally verifying the last of them) the authors conclude that the advantage of learning with quantum kernels is restricted to very particular functions.

**Limitations And Societal Impact:**

Limitations of this work: according to authors end of section 4 and section 6. However to me especially section 6 sounds more like limitations of quantum machine learning based on the results obtained. In any case I find it very good that authors mention and highlight that in practice the kernel would need to be determined via measurements, which is not something they consider in their analysis or experiments.

**Main Review:**

The paper introduces relevant analysis for the quantum machine learning community, showcasing that the very popular idea of quantum advantage in reality is not easy to achieve. This is very significant, and in this sense I would like to see this paper accepted.

While for the most part the authors have made admirable effort to as simply as possible communicate the complicated contributions (e.g. introduced theorems and concepts are thoroughly discussed), this is not fully successful. Some details in the paper remain unclear to me (at the level of mainly focusing on the main paper but also taking quick looks to appendix trying to see if it would clear things up), but I would assume mostly due to undefined notation. However not being able to understand the theorem statements etc fully hinders the full understanding the paper, and as it is difficult to fully understand I'm slightly reserved in my acceptance recommendation.


While I'm somewhat familiar with the concepts of quantum computation in machine learning, I am not very familiar with most of the literature there. Thus I cannot too accurately judge the novelty of the work and its relation to other quantum machine learning papers.


Comments and questions:

- You at the beginning of your paper introduce the two measures to assess the quality of a kernel to learn a function. I was expecting to see them in your theoretical analysis, but it seems that instead they are only used in the experimental validation?
- A comment about the measures to judge the quality of a kernel:  while having a high value on kernel target alignment to ideal kernel certainly indicates the good quality to learn f, it is very much possible that other kinds of kernels could be also reasonably good at the task even if they don't have a high alignment. On some easy learning problem RBF kernel can produce great results even if doesn't look like the ideal kernel. Could you comment if this has any impact on your analysis and conclusions?

- Line 228, what is \varphi? It has not been formally defined previously as far as I can see.

- Line between 251&252 Tr_1 has not yet been defined - is this partial or block trace? Does the last equality follow from some property detailed e.g. in [1]?

- Biased kernels: V just starts to appear (line 280) in superscript of \rho without any further introduction; only later it is said that it is a Haar-distributed random unitary matrix but it doesn't really explain why they are used with \rho. Nor does section 3 introduce such notation. Paragraph starting from line 227 introduces unitary tranformation V for the embeddings, but the notation in superscript is not used there either. Note: also Figure 1 uses this notation.

- Theorem 1:
  * line 266: what is k here? k has been the kernel function throughout the paper, now it's in the exponent.
  * for all d >= d0 no function can be learned with KRR and d-qubit kernel. What if d is smaller than d0, what happens then?
    -> it seems that you do not illustrate this theorem experimentally. While I realise that theorem 2 is more relevant to be illustrated mainly due to it not having a fixed V, it could be very illuminating to the reader to illustrate also this theorem.
  * what is the formula for d_0? I did not see this laid out in appendix (and the proof is overall difficult to grasp quickly as the parts introduced in theorem statement were not explicitly used there; to me it seems that the reader has to fill many gaps.)
  * Reading the theorem I can't help but think of deep neural networks, and other models with high number of parameters. Can we interpret that if d>=d0 then there are too many parameters for the restricted number of samples n for the function to be learned effectively? Or is my comparison completely faulty?

- Figure 2: gamma0 is stated on line 311 to be 0.5+O(2^-d); why is this constant in the figure when the x-axis displays d? Is it derived from your choice of keeping variances fixed?

- Abstract: "we show that finding suitable quantum kernels is not easy because the kernel evaluation might require exponentially many measurements" where is this shown? Something like this is discussed in paragraph "barren plateaus", but there it seems like kernel evaluation being costly is general knowledge. Or have I missed the assumption of requiring the measurements in some of the theorems?

- Might I suggest that for a reader not too familiar with quantum kernels but very familiar with kernel methods otherwise a table in which all the relevant things about quantum kernels (i.e. RKHS, kernel function, feature mapping...) are thoroughly detailed. (I realise space constraints probably won't allow this to be added to the main part of the paper.)




Minor comments (clear typos etc):
- Is "joint probability law" and "marignal law" (i.e. with the term law) standard language in this setting? Line 81 & 82. To me at least it would be more natural to use "distribution" rather than "law".
- First in line 95 K is used for kernel matrix; then in 101 and after that it is integral operator.
- typo on line 235: bet->be
- typo line 247 "consists of all functionS"
- "id" as identity matrix should be introduced as notation. Why not use I?
- Figures 2 & 3: left plot missing label for y-axis, Figure 2 left: it is not clear if the legend lists "gamma" or "y" (even if it can be deduced). Figure 2 right: are qw train and test on top of each other?
- Line 341, here "with high probability" is probably not an exact mathematical statement (but these words can be used as such in some settings).
- Line 346 "have to little bias" -> too little?
- Line 377 "Here" no need for capital H


[1] Filipiak et al, "THE PROPERTIES OF PARTIAL TRACE AND BLOCK TRACE OPERATORS OF PARTITIONED MATRICES", Electronic Journal of Linear Algebra, Volume 33, pp. 3-15

**Time Spent Reviewing:**

8

---

> ### Author Response · Authors · 2021-08-09
> **Response to Reviewer fXtY**
>
>
>
>
> Thank you very much for your positive review.
> We rewrote part of the Theorems and proofs to make things more accessible (see our comment to all reviewers, and response to X6Lb).
> Below we address your other comments and questions:
>
>
> 1. +2. *Kernel-target alignment and task-model alignment*: We introduce those quantities as qualitative/intuitive measures of when a kernel is good and use this to illustrate our experiments. As you point out in your second question they do not suffice to ultimately judge whether learning is feasible or not. Making a detailed theoretical connection between those measures is rather involved and beyond our scope (see Refs [32] and [33] for two recent publications). Our conclusions build on our Theorems, which work directly with the eigenvalues of the integral operator. Note that Equation (2) relates the kernel target alignment to the spectrum of the integral operator.
>
> 3. *l228* $\varphi_i$ is an arbitrary map that embeds a single coordinate into a qubit. We will clarify this.
>
> 4. *l251 &252*: Here we mean the partial trace over the first factor of the Tensor product space. We will clarify this. The equation follows from the linearity of the trace and the definition of partial trace (see Appendix A). We will include this into the Appendix.
>
> 5. We apologize and will be more explicit about the use of $V$ in superscript. Note that in 227ff we did not put $V$ as a superscript since $k(x,x')$ is invariant to $V$ due to the cyclic property of the trace and $V$ being unitary. However, this does not hold anymore for the biased kernels, which is why it appears here. We will make this clearer in the updated version.
> 6. *Theorem 1*: please see the general comment to all reviewers. Note that while there is no explicit formula for $d_0$, we now give simple conditions characterizing $d_0$. Regarding the analogy to neural nets: The analogy is correct for vanilla parametric linear models. In some cases, though, overparametrized models such as neural nets can also generalize. Here, it is not solely dimensionality that matters. For example, if the RKHS has a strong bias (i.e., "large" eigenvalues) for some functions, those can nevertheless be learned. Since in the Theorem the largest eigenvalue is exponentially small, *no* function has such a strong enough positive bias for $d > d_0$ to be learned with polynomial sample size. For small enough $d$, learning is successful, also with the quantum kernel. Note that Theorem 1 is implicitly illustrated in the right plot of Figure 2 where the orange curve shows the error of the full kernel which quickly grows.
> 7. *Figure 2*: First note that we expect that $\gamma_0$ is exponentially close to $1/2$ as $d$ grows. For our specific example (uniform distribution in each coordinate) the mean embedding is completely mixed (and thus also the mean embedding corresponding to the biased kernel) and thus $\gamma_0$ is exactly $0.5$.
> 8. *Barren Plateaus*: In practice we need the quantum device to estimate the kernel. We can only estimate $k(x,x')$ from binary measurement outcomes. To obtain sufficient precision we need exponentially (in $d$) many measurements to obtain a *single* entry of the kernel matrix. To our knowledge, our paper is the first that observes that one might need exponentially measurements to estimate every single kernel $k(x, x')$ accurately enough. Prior work only considers "Barren Plateaus" in the optimization landscape of Quantum Neural Networks.
> 9. We will include a table that summarizes the connection between feature map, quantum Hilbert space, kernel, and RKHS in the appendix.
>
>
> Re. you other points, we agree that 'distribution' will be the more common term used in machine learning. We will address the other points, too.

---

### Official Review · Reviewer_X6Lb · 2021-07-14

**Rating:** 6
**Confidence:** 5

**Summary:**

The paper presents the inductive bias of quantum kernels by analyzing their spectral properties. Specifically, their main theoretical results are that quantum kernels with too expressive RKHS will fail to generalize, and projected quantum kernels are a promising way to construct inductive biases that are hard to create classically.

**Limitations And Societal Impact:**

The results discussed in the submission can not be directly employed to solve practical learning tasks with provable quantum advantages.

**Main Review:**

Strengths:
This paper is well written. It provides a clear perspective to present some problems faced by quantum kernel methods through the lens of the inductive bias by analyzing the spectral of the corresponding integral operator.

Weakness:
My concerns lie in the following aspects about the two core theorems (Theorem 1 and 2):
-I am very confused about condition   $n \leq d^k$  in Theorem 1. First, I cannot find any clear definition of the notation "$k$". Is this a constant or a "kernel"? Second, in your proof, the usage of this condition seems to be unreasonable and even wrong. In particular, to ensure that the inequality in the bracket in the last row of the proof, the inequality "$\delta^{d/2} < \epsilon n^{-4}/2$" seems sufficient. Besides, the condition used in your proof is actually "$n>d^k$" instead of "`$n<d^k$" employed in Theorem 1.

-In the proof of Theorem 3 (the first line of page 23), it is unclear why $\lambda$ can be omitted in the derivation of inequality $\|\alpha\|^2 \le 2\|y\|^2$, as $K^{-1} \le 2I_{n}$ cannot deduce $(K+\lambda I_n)^{-1} \le K^{-1}$ when $\lambda$ can take any value. In the same place, the claim `for any $\epsilon > 0 $ with probability $1-\epsilon$ we have $\|y\|^2 \le \frac{n}{\epsilon}\|f\|_2^2$’ is not obvious.

-The conclusions and implications of Theorem 2 are arguable. Note that when $d$ grows, although the eigenfunctions are exponentially hard to compute classically, the eigenvalues and the projected kernel also exponentially tend to be a constant. Such a result can offset the quantum advantage brought by the eigenfunctions.

-In addition, there should be some specific explanations or appropriate references in the analysis of Theorem 2. For example, there should be an appropriate reference for the statement “However, …, but a proof would require the evaluation of 8-th order polynomials over the unitary group”.

Minor issues:

-the notation of kernel matrix and integral operator is the same.

-there is no necessary label of the y-axis in Figure 2 and Figure 3.

-the definition of $\| \cdot \|_{HS}$ is not clearly given.

-the subscript of $V_{\alpha, \alpha, j_1}$ in Eqn. (56) is confusing.

-the subscript of $T_{\alpha \beta, \gamma \delta}$ in Eqn. (62) is confusing.

-It should be $q_m^V(x, x’)=Tr[\widetilde{\rho}_m^V(x)  \widetilde{\rho}_m^V(x’)]$ in the introduction of `Biased kernels' in Page 8.

-It should be $\alpha=(K(X,X)+\lambda id_{n \times n})^{-1} y $ in the first line of page 23.


**Time Spent Reviewing:**

48h

---

> ### Author Response · Authors · 2021-08-09
> **Response to Reviewer X6Lb**
>
>
>
>
> We thank you for the substantial amount of time you spent on our paper. Your comments help us to improve the paper.
>
> Your concerns:
> 1. *Theorem 1 and its proof*: Please see the general comment to all reviewers.
>
> 2. *Theorem 3*: We assume that $\lambda \geq 0$ and although this is common usage for a regularisation parameter, this needs to be stated explicitly. Since $K$ is positive definite, for $\lambda \geq 0$ we indeed have $(K+\lambda I_n)^{-1} \leq K^{-1}$.
>
>     Next,
> we agree that the claim about the norm of $y$ is unnecessarily terse. The statement can be shown as follows: Note that $\mathbb{E}(y_i^2) = \mathbb{E}(f(x_i)^2) = \lVert f\rVert_2^2$ because we assumed that $x_i$ is i.i.d. with law $\mu$. Using Markov's inequality (and $|y|^2\geq 0$) we can bound $$\mathbb{P}\left(|y|^2\geq \frac{n}{\varepsilon} \lVert f\rVert_2^2\right) \leq \mathbb{E}\left(\frac{|y|^2}{n\varepsilon^{-1}\lVert f\rVert_2^2} \boldsymbol{1}[|y|^2\geq n\varepsilon^{-1}\lVert f\rVert_2^2 ] \right) \leq \frac{\varepsilon}{n\lVert f\rVert_2^2}\mathbb{E}\left(\sum_{i=1}^n |y_i|^2\right)=\varepsilon$$ where $\boldsymbol{1}[\cdot]$ denotes the indicator function. This implies that with probability at least $1-\varepsilon$
> the bound $|y|^2 \leq n\varepsilon^{-1}\lVert f\rVert_2^2$ holds.
>
> 3. *Conclusion of Theorem 2*: We agree with the reviewer. Indeed we discuss that our results can offset a quantum advantage, since the kernel is exponentially close to constant (see lines 317 - 324). We consider this "negative result" part of our contribution. We are not sure if this statement was meant as a criticism, and we would be happy to discuss.
>
> 4. We have shown that to bound the eigenvalues of $T_{\mu, m}^V$ we need to consider the tensor product
> \begin{align}
>     O_\mu = \int \tilde{\rho}_m^V(x) \otimes  \tilde{\rho}_m^V(x)\mu(\mathrm{d}x).
> \end{align}
> We calculate the expectation over $V$ in the paper. To further show that this holds with high probability over  $V$ we would need a bound for the variance of this expression. For this we need to evaluate the squared coordinates of $O_\mu$. This is an expression involving 4 terms $\tilde{\rho}_m^V$ which can be expanded as a polynomial of order 8=4+4 in $V$ and $V^\ast$. We agree that the statement was confusing and we will add a discussion in the supplement. We are happy to clarify any further issues in the discussion of Theorem 2.
>
> Thank you for your "minor issues". We will fix those in the revision.
>
> We hope that we adequately addressed your concerns and that you will now find the paper worthy of being presented at NeurIPS. Please indicate if more detailed clarification is required.

---

> > ### Comment · Reviewer_X6Lb · 2021-08-16
> > **Response to authors**
> >
> > The authors have well addressed my concerns. I would like to promote the rating score to 6. Note that the missing references about quantum kernels [1-3] should be added in the final version.
> >
> > [1] Wang, Xinbiao, et al. "Towards understanding the power of quantum kernels in the NISQ era." arXiv preprint arXiv:2103.16774 (2021).
> >
> > [2] Peters, Evan, et al. "Machine learning of high dimensional data on a noisy quantum processor." arXiv preprint arXiv:2101.09581 (2021).
> >
> > [3] Blank, Carsten, et al. "Quantum classifier with tailored quantum kernel." npj Quantum Information 6.1 (2020): 1-7.

---

### Official Review · Reviewer_J6gU · 2021-07-17

**Rating:** 7
**Confidence:** 2

**Summary:**

The paper gives a general investigation of quantum kernel learning, where quantum computers are used to compute the kernel functions used in standard kernel learning problems. Several results are given characterizing the general behavior of quantum kernels, which together give useful guidance towards the design of effective kernels for specific problems.

**Limitations And Societal Impact:**

Yes.

**Main Review:**

Given the recent interest in quantum machine learning, with quantum kernel learning being an important special case of this, the (mostly negative) results given here are useful for researchers in the field to identify promising areas for further investigation. The results themselves are a bit scattered, but they are applicable to general quantum kernel learning problems. These results mostly come from a spectral analysis of the quantum kernel, which is usefully related to properties of the underlying embedding of data into quantum density operators, and are clearly and thoroughly explained in the paper. All theoretical results are proved in full detail in the supplemental material.

A small collection of experiments are given which emphasize the utility of quantum kernels which are strongly biased towards specific problems, a fact which is justified by the theoretical results. The authors finally give a careful discussion about the impact of their results for further research in this area, which should be of interest to quantum ML researchers.

**Time Spent Reviewing:**

2

---

> ### Author Response · Authors · 2021-08-09
> **Response to Reviewer J6gU**
>
>
>
>
> Thank you for your positive assessment.
> We cannot find any explicit concerns in your review. Please let us know if we can provide further information that would allow you updating your confidence or score (from past experience, a score of 6 may not suffice for acceptance).

---

> > ### Comment · Reviewer_J6gU · 2021-08-18
> > **Response to the Authors**
> >
> > My apologies for not giving you anything concrete to address, I'm afraid to admit that the material was a bit more outside of my area of expertise than expected! But even with that, I do recognize the generality and rigor of the results, and appreciate your honesty in clearly identifying some of the difficulties faced by quantum kernel learning. After reading some of the more well-informed points made by the other reviewers, and in anticipation of some of the notational fixes in the revised version of the paper, I am increasing my score.

---

### Official Review · Reviewer_iZmY · 2021-07-18

**Rating:** 7
**Confidence:** 3

**Summary:**

This article studies the learnability of functions using quantum kernels. More precisely, it investigates the potential advantages in learning that quantum kernels may offer compared to classical ones. The focus was put on ridge kernel regression as a running example of a learning task. In this case, the analysis boils down to two ingredients: 1) the spectral properties of the corresponding integration operator 2) the alignment of the function to learn with the first eigenfunctions of the integration operator. The contribution of this work is to highlight the fact that, under some assumptions, the class of functions for which quantum kernels would have an advantage over classical ones is very restricted: the functions that belong to RKHSs associated with the so-called biased kernels.

**Limitations And Societal Impact:**

No.

**Main Review:**


Overall the article is well-written and tackles a crucial question in quantum learning theory. However, I believe that the conclusion is probably hasty since the analysis given in this work assumes that the kernel is separable:
$k(x,y) = \prod_{i \in [d]} k_i(x_i,y_i)$. For this class of kernels, the eigenvalues are the product of the eigenvalues of the respective uni-dimensional kernels: the eigenvalues of the corresponding integration operator are arranged by “plateaus”. It seems that this arrangement of the eigenvalues is crucial in the argument of this work (Section C.2. from line 629 to line 635 in the appendix). It would be interesting to study the case of a non-separable quantum kernel (Matérn kernels constitute a family of classic kernels that are non-separable without being pathological).



Minor observations:
- In line 266, the letter k is already used for the kernel k.
- In lines 717 and 718, I am not sure of the order in the inequalities since we assume that $n \leq d^{k}$.


**Time Spent Reviewing:**

9

---

> ### Author Response · Authors · 2021-08-09
> **Response to Reviewer iZmY**
>
>
>
> Thank you for your positive assessment. You raise some concerns regarding the applicability of our results for non-separable kernels.
> Let us emphasize that Lemma 1, Theorem 2, and Theorem 3 (in the Appendix) all do not require a separable kernel and hold more generally. We consider the specific separable kernel as an illustrative example (Note that 629-635 is not part of the proof) since it allows for a very clear construction when $d$ grows. Theorem 1 is indeed specific to this case, however, similar results can be derived from Theorem 3 for non-separable kernels whose largest eigenvalue decays exponentially as $d$ increases.
>
> We trust that this should address your concerns, and we would be happy to discuss further.

---

### Author Response · Authors · 2021-08-09
**Response to all Reviewers and AC**



Thank you all for the valuable reviews, pointing out that the paper "tackles a crucial question in quantum learning theory" (iZmY) and that "this is very significant" (fXtY). Overall, three reviewers favor acceptance. X6LB is less favorable due to concerns regarding the proofs of our Theorems, but generally finds that our paper provides a "clear perspective."

The main common criticism regards the statement of Theorem 1 and its proof. Therefore we discuss this in this general response while otherwise, we address the reviews individually.

We hope that with our below clarifications the reviewers are more confident in supporting acceptance.

Reviewer fXtY kindly wrote "While for the most part the authors have made admirable effort to as simply as possible communicate the complicated contributions (e.g. introduced theorems and concepts are thoroughly discussed), this is not fully successful." We realize that Theorem 1 was such a case, and we apologize. We will ensure that the final version, prepared without deadline pressure, will be readable throughout.



=================

(Below solely concerns Theorem 1)

As pointed out by iZmY, X6Lb, and fXtY the statement of the Theorem was confusing (double use of $k$ for kernel and an integer, implicit assumption that the regularization parameter $\lambda$ is non-negative). Furthermore, the proof was not detailed enough, and by an oversight we reported the wrong order of the inequalities in lines 717-718 (Reviewer iZmY politely wrote "I am not sure of the order in the inequalities"; this was also noticed by Reviewer X6Lb).

We are thankful to the reviewers for pointing this out and apologize for any inconvenience caused (we will acknowledge the anonymous reviewers in the final version). Let us emphasize that the Theorem itself is correct, and the changes to the proof are only to (1) fix the above order of the inequalities and two typos, and (2) include more detail.


We clarify the Theorem and expand the proof as follows. Please let us know if further clarification is required.

**Theorem 1.** Suppose $\mu=\bigotimes \mu_i$ and that the purity of the embeddings $\rho_{\mu_i}$ satisfies $\text{Tr}[{\rho_{\mu_i}^2}]\leq \delta < 1$ as the dimension and number of qubits $d$ grows. Furthermore, suppose the training sample size only grows polynomially in $d$, i.e., $n\leq d^l$ for some fixed $l\in \mathbb{N}$. Then there exists $d_0 = d_0(\delta, l, \varepsilon)$ such that for all $d \geq d_0$
no function can be learned using kernel ridge regression with the
$d$-qubit  kernel $k(x,x')=\text{Tr}[{\rho(x)\rho(x')}]$
in the sense that for any $f \in L^2$, with probability at least $1-\varepsilon$ for all $\lambda\geq 0$
$$
    R(\hat{f}^\lambda_n)\geq (1-\varepsilon) \lVert f\rVert^2.
$$

**Proof of Theorem 1** The general strategy of the proof is to show that the result
follows from
Theorem 3 for sufficiently large $d$.
We first note that, using the assumption
$\mu = \bigotimes \mu_i$
$$
    \rho_\mu = \bigotimes \rho_{\mu_i}
$$
and thus
$$
    Tr{\rho_\mu^2} = \prod Tr{\rho_{\mu_i}^2} \leq \delta^d.
$$
Lemma 1 then implies that the largest eigenvalue of the integral operator is bounded by $\gamma_{max}(d)\leq \delta^{d/2}$.
Next we observe that there is $d_0(\delta, l, \varepsilon)$
such that for $d\geq d_0$
$$
\delta^{d/2} \leq \varepsilon d^{-4l}/2 \qquad \text{and} \qquad \delta^{d/2} \leq \varepsilon^3 d^{-2l}/4,
$$
because the left hand sides of the equations are decaying exponentially in $d$ (recall that $\delta < 1$) and the right hand sides only polynomially.

Using the estimates above and the assumption $n\leq d^l$ we conclude that for $d\geq d_0$
$$
    \gamma_{max}\leq \delta^{d/2} \leq \varepsilon d^{-4l}/2
    \leq \varepsilon {n^{-4}}/{2}
    \quad \Rightarrow \gamma_{max}n^4\leq \varepsilon/2
$$
and
$$
    \gamma_{max}\leq \delta^{d/2} \leq \varepsilon^3 d^{-2l}/4
    \leq \varepsilon^3 {n^{-2}}/{4}
    \quad \Rightarrow \sqrt{4\gamma_{max}n^2\varepsilon^{-1}}\leq \varepsilon.
$$


We now denote the $\varepsilon$ used in Theorem 3 as $\varepsilon'$ and set $\varepsilon'=\varepsilon / 2$.
Theorem 3 and the bounds above then imply that with probability at least
$$
    1 - \varepsilon' - \gamma_{max}n^4
    \geq 1 - \varepsilon / 2 - \varepsilon /2
    = 1 - \varepsilon
$$
the bound
$$
    \lVert f - \hat{f}^\lambda_n \rVert_2^2
    \geq \left( 1 - \sqrt{\frac{2\gamma_{max}n^2}{\varepsilon'}}\right)\lVert f\rVert_2
     = \left( 1 - \sqrt{\frac{4\gamma_{max}n^2}{\varepsilon}}\right)\lVert f\rVert_2
    \geq (1-\varepsilon)\lVert f\rVert_2
$$
holds for all $\lambda\geq 0$. This completes the proof.

---

### Decision · Program_Chairs · 2021-09-27

**Decision:**

Accept (Poster)

**Comment:**

There is a consensus among the reviewers that this is a very strong theoretical analysis for the learnability of functions using quantum kernels. The results discovered poor learnability of quantum kernels which is similar to the “barren plateau” phenomenon in quantum neural networks and provided useful guidance towards the design of effective kernels for specific problems. In summary, the paper is helpful for understanding the power of quantum kernels, and I would like to recommend to accept this submission.